



# Evaluation of debris-flow building damage forecasts

Katherine R. Barnhart[1], Christopher R. Miller[2], Francis K. Rengers[1], and Jason W. Kean[1]

1 U.S. Geological Survey, Geologic Hazards Science Center, Box 25046 DFC MS 966, Denver, CO 80225-0046, USA
5  2 Independent researcher, Denver, CO, USA

*Correspondence to*: Katherine R. Barnhart (krbarnhart@usgs.gov)

## Abstract

Reliable forecasts of building damage due to debris flows may provide situational awareness and guide land and emergency
10   management decisions. Application of debris-flow runout models to generate such forecasts requires combining hazard
intensity predictions with fragility functions that link hazard intensity with building damage. In this study, we evaluated the
performance of building damage forecasts for the 9 January 2018 Montecito postfire debris-flow runout event, in which over
500 buildings were damaged. We constructed forecasts using either peak debris-flow depth or volume flux as the hazard
intensity measure and applied each approach using three debris-flow runout models (RAMMS, FLO-2D, and D-Claw).
Generated forecasts were based on combining multiple simulations that sampled a range of debris-flow volume and mobility,
reflecting typical sources and magnitude of pre-event uncertainty. We found that only forecasts made with volume flux and
the D-Claw model could correctly forecast the observed number of damaged buildings and the spatial patterns of building
damage. However, the best forecast only predicted 50% of the observed damaged buildings correctly and had coherent spatial
patterns of incorrectly forecast building damage (i.e., false positives and false negatives). These results indicate that forecasts
made at the building level reliably reflect the spatial pattern of damage, but do not support interpretation at the individual
building level. We found the event size strongly influences the number of damaged buildings and the spatial pattern of debris-
flow depth and velocity. Consequently, future research on the link between precipitation and the volume of sediment mobilized
may have the greatest effect on reducing uncertainty in building damage forecasts. Finally, because we found that both depth
and velocity are needed to forecast building damage, comparing debris flow models against spatially distributed observations
of building damage is a more stringent test for model fidelity than comparison against the extent of debris-flow runout.

## 1 Introduction

Debris flows are sediment and debris-laden flows that may initiate from shallow landslides or overland flow runoff (Cannon,
2001; Iverson, 1997). Buildings, roads, bridges, and other infrastructure located downstream from catchments susceptible to
debris flows are exposed to this hazard. Debris flows pose a hazard to buildings that can result in damage ranging from slight
(e.g., failure of non-load bearing components) to complete destruction (e.g., substantial structural damage, removed from



foundation) (Jakob et al., 2012). A reliable approach to forecast building damage in areas susceptible to debris-flow runout would be useful for multiple decision-making activities, such as evacuation planning (Barnhart et al., 2023).

A fragility function relates a measure of hazard intensity (e.g., debris-flow depth, tsunami velocity, or peak ground acceleration from an earthquake) to the corresponding likelihood of a specific type of asset (e.g., a building) meeting or exceeding a categorical damage state. The development of fragility functions for specific asset types and specific hazards is an established field (e.g., Baker et al., 2021; FEMA, 2022a). Multiple types of fragility functions exist, including empirical fragility functions based on inventories of damaged assets, analytical fragility functions based on physics or engineering first principles, and expert elicitation-based methods. Examples of proposed measures of hazard intensity for empirical or analytical debris-flow fragility functions include the following: debris-flow depth (Fuchs et al., 2007), the ratio of debris-flow depth to building height (Totschnig et al., 2011), the volume flux (product of debris-flow depth and velocity squared, also called the momentum flux or the impact force; Jakob et al., 2012), the overturning moment (product of depth and velocity; Zhang et al., 2018), and the impact pressure (product of density and velocity squared; Calvo and Savi, 2009).

The objective of this contribution was to evaluate the performance of building damage forecasts generated by combining runout-model output with a fragility function. We were interested in understanding the performance of building damage forecasts in locations with limited information about past debris-flow runout activity (e.g., recently burned areas). This type of application is distinct from evaluation of building damage potential in areas with a historical record of debris flows that may be used to back calculate model parameters (e.g., Quan Luna et al., 2011). Should it be possible to construct a reliable building damage forecast based on probabilistic sampling of runout model input parameters, such a methodology may be more widely applicable than one based on calibrated parameters.

Runout models simulate the dynamic evolution of debris-flow material as it moves across the landscape under the force of gravity. Thus, the output of runout models (i.e., debris-flow depth, velocity) can be used as the input to a fragility function to forecast building damage. Prior studies have used runout models to generate fragility functions (e.g., Zhang et al., 2018) and evaluate building failure modes (e.g., Luo et al., 2022), but few studies have evaluated the performance of building damage forecasts generated by combining preexisting fragility functions with the output of uncalibrated runout models in the context of an observed event. Accordingly, there are many unanswered questions surrounding how to apply runout models to construct forecasts of building damage. These include (1) Which fragility functions, runout models, and measures of hazard intensity produce the most reliable forecasts? (2) How should uncertainty in debris-flow size and mobility be combined to generate probabilistic forecasts of building damage? and (3) What level of performance and spatial specificity can be expected for building damage forecasts?





To accomplish this objective, we developed a method for constructing probabilistic building damage forecasts and applied it
to the 9 January 2018 Montecito, California, debris-flow event (Kean et al., 2019b; Lancaster et al., 2021; Oakley et al., 2018)
(hereafter "Montecito event"). This event damaged over 500 primarily wood-framed buildings (Lancaster et al., 2021;
Lukashov et al., 2019), thereby providing a spatially distributed dataset of building damage. The method we propose is general
because it can be used with different runout models, different hazard intensities, and different fragility functions. We evaluated
the relative performance of two hazard intensity measures (debris flow depth and volume flux) and three different runout
models (RAMMS, FLO-2D, and D-Claw, Christen et al., 2010; George and Iverson, 2014; Iverson and George, 2014; O'Brien
et al., 1993; O'Brien, 2020). We considered five event size categories ranging from much smaller to much larger than the
observed event. Within each combination of model and event size category, we combined multiple simulations that reflect the
pre-event uncertainty in event size. Our goal was not to comprehensively test all available runout models, hazard intensities,
or fragility functions, but instead to evaluate approaches that vary in their complexity.

We evaluated the forecasts using standard methods of forecast validation developed in the atmospheric sciences. For the best
performing model and the best performing hazard intensity measure, we performed two follow-on analyses. First, we examined
the sensitivity of the simulated hazard intensity to model inputs, which indicated where further research may be most effective
at reducing pre-event uncertainty in building damage forecasts. Finally, we estimated the minimum number of simulations
required to generate statistically equivalent results.

The remainder of this contribution is organized as follows: In sections 2 and 3 we describe the Montecito event, the building
damage dataset, and a previously developed set of runout model simulations. We then propose our method to generate a
probabilistic forecast of building damage (section 4.1). This method requires a fragility function, and we introduce two
candidate approaches in section 4.2. In the remainder of section 4 we describe our approach to forecast evaluation, how we
evaluate the sensitivity of forecasts to model input, and how we determine the minimum number of simulations needed to
produce similar results.

Our results document three main findings: Forecasts generated with D-Claw and using a fragility function based on debris
flow volume flux outperform all other approaches (section 5.2). The total volume of mobilized sediment and water, which we
refer to as the event size, is the most important model input, influencing the number of buildings damaged and the spatial
pattern of which buildings are damaged (section 6.1.2). Finally, the forecast evaluation identities systematic errors that may
indicate priority areas for fundamental model improvement (section 6.2.2).

## 2 Event description



Our study focused on the 9 January 2018 Montecito, California, debris-flow event (hereafter "Montecito event") (Kean et al., 2019b; Lancaster et al., 2021; Oakley et al., 2018). This event was initiated by intense rain (5-minute intensity of 157 mm.hr$^{-1}$) that fell on the recently burned Santa Ynez Mountains. The event mobilized sediment from hillslopes and channels (Alessio et al., 2021; Morell et al., 2021) into a boulder-laden slurry that ran out onto a ~4 km-wide alluvial fan located between the Santa Ynez Mountains and the Pacific Ocean (Figure 1). The debris-flow runout inundated a combined area of 2.6 km$^2$ and

resulted in 23 fatalities, at least 167 injuries, and over five hundred damaged homes (Lancaster et al., 2021; Lukashov et al., 2019).

Prior work estimated the total amount of sediment deposited in the event (Kean et al., 2019b), eroded from the hillslopes (Alessio et al., 2021), and eroded from the channels (Morell et al., 2021). Barnhart et al. (2021) combined the sediment volumes

estimated by Kean et al. (2019b) upstream from three domains with an estimate of water volume based on rainfall-runoff analysis to produce an estimate of the total event size (volume of water and sediment) for each domain: 531,000 m$^3$ for Montecito Creek, 522,000 m$^3$ for San Ysidro Creek, and 332,000 m$^3$ for Romero Creek. Barnhart et al. (2021) considered an arbitrary factor of two uncertainty estimate for the event volume (50%–200%). Because more recent work by Alessio et al., (2021) and Morell et al. (2021) found the total volume of sediment eroded from hillslopes and channels during the event

matched the estimates of deposit volume, here we considered a smaller, although still arbitrary, uncertainty range of 70%–130% on these volumes.

## 3 Data

Generation and evaluation of building damage forecasts required a dataset of the location and damage state of buildings in Montecito, California, and simulation output of spatially distributed values of peak flow depth, h (m) and volume flux hv$^2$ (m$^3$

s$^{-2}$) . Here, v (m s$^{-1}$) is the flow velocity. Generation of a candidate fragility function required observed damage state and observed flow depth. This section describes the data sources used in our analysis.

### 3.1 Building dataset

After the Montecito event, building inspectors produced a database of damaged homes that was compiled with observed debris-flow characteristics and published by Kean et al. (2019a). Initial observations were generated by the California Department of

Forestry and Fire Protection (CAL FIRE) building inspectors who classified impacted buildings into four ordered damage class categories: affected, minor damage, major damage, and destroyed following the categories described by the Federal Emergency Management Agency (FEMA) Preliminary Damage Assessment Guide (FEMA, 2021) (examples of building damage depicted in Figure 2). We note that this damage classification scheme is neither strictly economic nor strictly structural. Additionally, in the dataset disseminated by Kean et al. (2019a), these four categories were labeled 1%–9% damaged, 10%–25% damaged,

51%–75% damaged, and destroyed, respectively. Kean et al. (2019b) supplemented these damage class observations with



observed debris-flow depth and building attributes (area and width of building footprint, number of stories, and age of buildings).

To calculate the number of buildings simulated as damaged, we needed information describing the location of all buildings in the Montecito area that were not damaged by the 2018 event because a simulation might predict that debris flow runout would affect an area that was not affected by the observed event. Therefore, we supplemented this database of observed building damage with the location of all undamaged buildings in the considered simulation domains from OpenStreetMap (OSM, https://www.openstreetmap.org/, database accessed November 12, 2021) (Figure 1). We removed any OSM-sourced buildings that overlapped with a building in the CAL FIRE dataset to prevent duplication. The OSM-sourced buildings were categorized

as unimpacted, yielding a total of five damage categories. The final dataset contained 4002 unimpacted buildings, 127 buildings with 1%–9% damage, 126 buildings with 10%–25% damage, 114 with buildings 51%–75% damage, and 162 destroyed buildings (Table S1).

We simplified the building damage dataset from the five original categories to two categories separating major and minor damage. We refer to this simplified damage category as $D_s$. Buildings classified as unimpacted, affected, and minor damage

are all associated with $D_s=0$, whereas major damage and destroyed are associated with $D_s=1$. The boundary between minor and major damage corresponds with the difference between repairable, non-structural damage to substantial or structural damage (FEMA, 2021). We chose to simplify the damage categories at the boundary between minor and major damage because it is most consistent with the needs of emergency managers: to identify areas where debris-flow runout poses a threat to life

and property (Barnhart et al., 2023).

## 3.2 Simulated event size, flow depth, and volume flux

We used simulation results from a prior study (Barnhart et al., 2021) that evaluated the ability of three different runout models [RAMMS (Christen et al., 2010), FLO-2D (O'Brien et al., 1993; O'Brien, 2020), and D-Claw (George and Iverson, 2014; Iverson and George, 2014)] to match the extent of debris-flow runout. These authors ran multiple simulations with each model

and in this section, we describe their sampling strategy, how peak flow depth and volume flux were extracted from the simulations, and how each simulation was categorized based on event size.

Barnhart et al. (2021) used a Latin hypercube sampling study to generate parameter values for each simulation. All models used the event size, specified as the debris-flow volumes, $V$ (m$^3$). Each model used a different set of governing equations and, thus, a different set of inputs that describe the mobility of debris-flow material. For a given model, the number of simulations

was determined as $100\times$ the number of model free parameters, $N_p$ ($N_p = 3$, 5, and 4 for RAMMS, FLO-2D, and D-Claw,





respectively, as described by Barnhart et al. (2021). Finally, Barnhart et al. (2021) split up the complex runout path from the Montecito event into three separate simulation domains (Figure 1).

Simulations were conducted on a 5-m bare-earth digital elevation model and consequently, the simulated values of debris-flow depth and velocity represent the values without explicit representation of the interaction between the flow and the building. For each simulation, the maximum debris-flow depth, h, and maximum volume flux, $hv^2$, was recorded at each grid cell (5-m cell sides). For each of the simulations presented in Barnhart et al. (2021), we extracted the maximum simulated debris-flow depth and volume flux at the model grid cell containing the centroid of every considered building. Files compiling the

maximum h and $hv^2$ for each simulation at each building are provided in the data release associated with this contribution (Barnhart et al., 2023).

One objective of our study was to understand how uncertainty in pre-event unknowns, such as the rainfall intensity and associated debris-flow volume, propagate into a forecast of building damage. Therefore, we designed our approach to generate

forecasts based on forecast rainfall. We were able to accomplish this objective because prior work has established a link between the 15-minute rainfall intensity, $I_{15}$, and the mobilized volume (Gartner et al., 2014). Barnhart et al. (2021) used the volume of water that would fall on each catchment in 15 minutes given a specified rainfall intensity ($I_{15}$) and the volume of sediment used by the current U.S. Geological Survey emergency hazard assessment methodology (Gartner et al., 2014; U.S. Geological Survey, 2018). The underlying statistical model used in the emergency assessments to predict mobilized sediment

volume has a sub-linear relation between the natural logarithm of sediment volume and $I_{15}$ (Figure S1). However, this sub-linear fit has nearly an order of magnitude prediction uncertainty (Gartner et al., 2014). Accordingly, one of the most uncertain aspects of forecasting the hazard of postfire debris flows is the link between forecast rainfall, as represented by $I_{15}$, and the expected event size, as represented by the total volume of sediment and water.

Barnhart et al. (2021) generated event volumes that ranged from less than four times smaller to more than four times larger than the observed event size, and we split the simulations done by Barnhart et al. (2021) into five groups based on the simulated event size and generated forecasts with each model for each event size. Forecasts generated with simulations that had an event volume similar to what was observed in the Montecito event (section 2) are referred to as having an unbiased event magnitude. Forecasts generated with simulations that had event volumes smaller than the observed event are referred to as having an

underforecast or very underforecast event magnitude. Forecasts generated with simulations that had event volumes larger than the observed event are referred to as having an overforecast or very overforecast event magnitude. The volume ranges within each event magnitude category and number of simulations vary by domain (Table S2). The volume values used to split simulations into the five groups were informed by the observed event size and the prediction uncertainty associated with forecasting event size based on the $I_{15}$ (Figure S1).






## 4 Methods

We generated probabilistic building damage forecasts using multiple models and fragility function methods. In this section we describe a general approach to generating probabilistic building damage forecasts based on model output and two approaches to constructing fragility functions: (1) the approach used here to implement the general approach in the context of the Montecito

event, and (2) the methods used to evaluate forecast performance (Figure 3). Results motivated two follow-on analyses: (1) How sensitive are forecasts to each model input parameter?; and (2) How few simulations are needed to generate similar results to those presented here? The first of these questions is relevant for identifying what observations may be most important for reducing forecast uncertainty, and the second has practical importance for generating similar results with limited time or computational resources.

### 4.1 General method for probabilistic construction of building damage forecasts

We used a simple and general method for constructing probabilistic forecasts of building damage: combining the results of multiple simulations and weighting them equally. Consider a set of N simulations generated by sampling input parameter values such that the set of simulations reflects pre-event uncertainty. Assuming that output from simulation i at building $x_b$ can be transformed into the probability (P) that $D_s=1$, the probability that $D_s=1$ across all simulations is given as:

$$P(D_s = 1|X = x_b) = \frac{1}{N}\sum_{i=1}^{N}[P(D_s = 1| X = x_b)]_i \tag{1}$$

where:

$x_b$ is the unique identifier for each building X,

N is the number of simulations being combined, and

$[P(D_s = 1 | X=x_b)]_i$ is the probability that $D_s=1$ for building $x_b$ in simulation i.

Equation (1) can be interpreted as equally weighting the likelihood of each simulation and taking an average. In our application,

the N simulations each use a different value for event size and flow mobility, but other applications may evaluate other sources of pre-event uncertainty.

### 4.2 Fragility functions

To generate $P(D_s = 1 | X=x_b)$ for use in Equation (1), we used a fragility function that transforms a measure of hazard intensity into the probability of damage. We considered two fragility functions, the first is an empirical fragility function specific to

wood-frame buildings that was derived based on observed peak flow depths from the Montecito event, and the second uses an existing methodology developed for tsunami hazard assessment based on volume flux.





### 4.2.1 Empirical fragility function using peak depth

Because $D_s$ is a binary variable, we used logistic regression to predict $D_s$ with $\ln(h)$. We fit the following equation with the observed values of $\ln(h)$ and $D_s$ using the generalized linear models (glm) function provided by the core stats package in R (R

Core Team, 2022):

$$P(D_s = 1|\ln(h)) \ = \ \Phi(\beta_0 + \beta_1\ln(h)) \tag{2}$$

where:

   $\beta_0$ and $\beta_1$ are estimated constants, and

   $\Phi(\cdot)$ is the cumulative standard normal distribution function.

Given a hazard intensity, application of equation (2) yields a predicted probability that $D_s=1$, and the building will have major

damage or be destroyed. To classify each prediction into the discrete ordinal values of 0 and 1, a discrimination threshold, or a cut point probability value, is typically used. We determined the discrimination threshold for classification by evaluating how the standard binary classification metrics bias and threat score varied as a function of discrimination threshold. We selected the discrimination threshold as the probability value that maximized the threat score and had a bias close to unity (definitions of these metrics provided in section 4.4). We used this method rather than a receiver operating characteristic curve

analysis because the underlying observation data includes many undamaged buildings that experienced no damage and was thus unbalanced.

Because we used the observations of building damage dataset to generate the empirical fragility function and these same observations were used to evaluate simulation results, we comment here on whether this choice adds any circularity into our

method. One might be concerned with circularity because the same building data being used to train the empirical fragility function described in this section are used to test the runout model forecasts. However, because the building data are being used in two different ways with two independent sets of debris-flow depths, our use is not circular. To generate the empirical fragility function, we used the building damage data alongside observations of debris-flow depth to generate a relation between depth and likelihood of damage. Later we evaluate the ability of a runout model to predict the spatial pattern of building

damage based on simulated debris-flow depths. Because the runout models were not calibrated to match the building damage observations, the use of observed damage to both generate an empirical fragility function and evaluate the results is not circular.

### 4.2.2 Hazus fragility function using peak volume flux

We also forecast building damage based on the peak volume flux, $hv^2$, by applying the Hazus methodology for "Building damage functions due to tsunami flow" (FEMA, 2022a, p.5–22). The Hazus model determines building damage class by

comparing the magnitude of the debris-flow impact force, $F_{DF}$ (kg m s$^{-2}$), and the lateral strength of the building. $F_{DF}$ is a building-specific value that is calculated based on the drag equation (i.e. equation 5.36 in Furbish, 1997):



$$F_{DF} = \frac{1}{2} K_D \rho C_D B_W \overline{hv^2} \tag{3}$$

where:

$K_D$ (unitless) accounts for uncertainty in loading,

$\rho$ (kg m$^{-3}$) is the density of the flow,

$C_D$ (unitless) is the drag coefficient,

$B_W$ (m) is the width of the building perpendicular to the flow direction, and

$\overline{hv^2}$ is the median volume flux.

Following the Hazus methodology for estimating building damage based on tsunami flow, we estimated $\overline{hv^2}$ as 2/3 of the peak volume flux (FEMA, 2022a, p.4–18). The probability of $D_s=1$, given a value for $\overline{hv^2}$, is given by a lognormal distribution

$$P(D_s = 1|\overline{hv^2}) = \Phi\left(\frac{1}{\beta_j} \ln \frac{\overline{hv^2}}{\zeta}\right) \tag{4}$$

where:

$\beta_j$, is the lognormal standard deviation associated with damage class $D_s=1$,

$\zeta$, is the median value of the volume flux (m$^3$ s$^{-2}$) associated with damage class $D_s=1$, and

$\Phi(\cdot)$ is the cumulative standard normal distribution function.

The value for $\zeta$ is given by substituting $\zeta$ for $\overline{hv^2}$ in Equation (3), equating $F_{DF}$ with a critical force per unit area, $F_C$ (kg m s$^{-2}$), and rearranging for $\overline{hv^2}$.

$$\frac{1}{2} K_D \rho C_D B \zeta = F_C \tag{5}$$

$$\zeta = \frac{2F_C}{K_D \rho C_D B_w} \tag{6}$$

Here we have followed Kean (2019b) in calculating $F_C$ as the mean of the yield and ultimate pushover strengths, $F_Y$ and $F_U$, respectively. These two values are calculated individually for each building:

$$F_Y = \alpha_1 A_Y W \tag{7}$$

$$F_U = \alpha_1 A_U W \tag{8}$$

where:

$\alpha_1$ is the modal mass parameter,

$A_Y$ is the fraction of gravitational acceleration at yield,

$A_U$ is the fraction of gravitational acceleration at pushover, and

W is the total building seismic design weight (FEMA, 2022a, p.5-27, equations 5.12 and 5.13).



To calculate $F_Y$ and $F_U$, building attributes such as Hazus building type (e.g., W1 and W2 for wood frame), age, and number of stories must be known. For the purposes of this analysis, we assumed that all buildings are one story wood frame buildings built between 1941 and 1975 (buildings built with the same seismic design level). We acknowledge this is a simplification, but it matches the character of residential buildings damaged in this event (Table S3 and Table S4). We discuss the implications of these simplifications later in the text. $A_U$ and $A_Y$ are typically calculated based on building characteristics found in Table

5.7 from the Hazus Earthquake Model Technical Manual (FEMA, 2022b). Accordingly, we used $\alpha_1$ =0.75, $A_Y$ = 0.3, and $A_U$ =0.9 for buildings with area less than 465 m$^2$, and $A_Y$ = 0.2 and $A_U$ =0.5 for buildings with footprint area greater than 465 m$^2$. Following Kean et al. (2019b) we calculated W using the footprint area and a value of 1820 N m$^{-2}$ for the structural weight per area and a value of $\beta_j$=0.633 for all damage classes. We used a debris-flow density of 2020 kg m$^{-3}$ reflecting a weighted average of water (1000 kg m$^{-3}$) and sediment (2700 kg m$^{-3}$) and a solid volume concentration of 0.6.

**4.3 Forecast construction**

For each model (RAMMS, FLO-2D, D-Claw) and event magnitude forecast bias classification (five categories) we constructed a building damage forecast using h and the empirical fragility function (Equation (2)) and another using hv$^2$ and the Hazus methodology (Equation (4)). Each of these 30 forecasts provides a probability that the simplified damage category, Ds, introduced in section 3.1, at each building is equal to 1, indicating the building would experience major damage or be destroyed.


Each forecast combined the results of multiple simulations using Equation (1). The simulations used to generate each forecast reflect typical pre-event uncertainty in debris-flow mobility and event size, within the range of volume for the event magnitude forecast bias category. Accordingly, the probability of building damage in a specific forecast reflects uncertainty associated with event size and mobility. Comparison between the forecasts made with different event magnitude forecast categories

documents the sensitivity of forecast performance to getting the event size approximately correct (within a factor of two). We generated example forecasts for multiple event magnitude forecast bias categories for two reasons: (1) the forecast event size is characterized by considerable uncertainty, even if forecast rainfall is well known, and (2) event rainfall is itself difficult to predict (Gartner et al., 2014; Oakley et al., 2023).

**4.4 Forecast evaluation**

Each forecast provides a probability value for each building, and we evaluated the forecasts based on the spatial pattern of predicted building damage and aggregated measures of performance. We classified buildings with a probabilistic damage forecast of 50% or greater as having predicted damage and then calculated the four elements of the binary classification contingency table for each forecast. Buildings with observed and predicted damage were classified as true positive (TP); buildings with predicted but not observed damage were classified as false positive (FP); buildings with observed but not

predicted damage were classified as false negative (FN); and buildings with neither observed nor predicted damage were





classified as true negative (TN). Using TP, FP, and FN, we calculated three standard summary values: the false alarm ratio (FAR), hit rate (H), threat score (TS), and bias (B).

$$FAR = \frac{FP}{TP+FP} \tag{9}$$

$$H = \frac{TP}{TP+FN} \tag{10}$$


$$TS = \frac{TP}{TP+FP+FN} \tag{11}$$

$$B = \frac{TP+FP}{TP+FN} \tag{12}$$

The number of buildings with observed damage is TP+FN, and the number of buildings with forecast damage is TP+FP. Thus, FAR, H, TS, and B are interpreted as follows:

- FAR is the fraction of buildings forecast as damaged that were not damaged in the debris flows. If FAR is equal to
zero, no false positive predictions were made.
- H is the fraction of buildings observed as damaged that were forecast correctly. If H is equal to one, no false negative predictions were made.
- TS is the proportion of correct forecasts, disregarding the TN category. If TS is equal to one, no false positive or false negative predictions were made.
- B is the ratio of number of forecast damaged buildings and observed damaged buildings. If B is equal to one, the same number of buildings observed as damaged are predicted as damaged. However, B=1 does not guarantee that the correct buildings are forecast as damaged; that would require both B and TS be equal to one.

We compared all forecasts using a commonly used graphical method for forecast verification developed in the atmospheric sciences called the Roebber (2009) performance diagram (Wilks, 2019, p.384). The Roebber (2009) performance diagram
plots (1–FAR) on the x-axis and H on the y-axis and is best used for comparing forecasts of rare events or events for which the number of TN is unconstrained. In the application presented here, the value of TN is arbitrarily set by the extent of the simulation domains. A convenient property of the Roebber (2009) performance diagram is that it can be contoured with isolines of constant TS and B such that changes in H, FAR, TS, and B can all be evaluated simultaneously.

## 4.5 Sensitivity of hazard intensity to model input

Because the results of the forecast evaluation (presented in section 5.2) indicated that D-Claw was the best performing model for predicting building damage, we wanted to understand the sensitivity of the two model outputs, h and v, to each of the D-



Claw input parameters. This analysis was done to document which of the input parameters was most important in generating variability in the model outputs. Input parameters with high importance have greater impact on the simulated outputs such that reducing pre-event uncertainty in those parameters will have the greatest effect on reducing uncertainty in the building damage
forecasts.

At every building, we used the results of all considered simulations to evaluate the ability of each D-Claw input parameter to predict h and v, the two elements of $hv^2$ by fitting a linear model of the following form:

$$\frac{y}{\sigma_y} = c_0 + c_1 \frac{\log_{10}(V)}{\sigma_V} + c_2 \frac{m'}{\sigma_{m'}} + c_3 \frac{\log_{10}(k')}{\sigma_{k'}} + c_4 \frac{\phi_{bed}}{\sigma_\phi} \tag{13}$$

where:

    y is the output of interest, h or v,
$\sigma_y$ is the standard deviation of y,

    $c_0$, $c_1$, $c_2$, $c_3$, and $c_4$ are estimated regression coefficients,

    $\log_{10}$ (V) is the base-10 logarithm of the total event volume,

    $\sigma_V$ is the standard deviation of $\log_{10}$ (V),

    m′ is the difference between the initial solid volume fraction and the critical solid volume fraction,
$\sigma_{m'}$ is the standard deviation of m′,

    $\log_{10}$ (k′) is the base-10 logarithm of the ratio of the timescale of downslope debris motion and the relaxation of pore pressure,

    $\sigma_{k'}$ is the standard deviation of $\log_{10}$ (k′)

    $\phi_{bed}$ is the basal friction angle, and
$\sigma_\phi$ is the standard deviation of $\phi_{bed}$.

The four model input parameters—log (V), m′, log (k′), $\phi_{bed}$—were normalized by dividing by their standard deviations before fitting the regressions to make the coefficient values comparable. Similarly, h and v were rescaled by dividing by their standard deviations.

The relative magnitude of each regression coefficient indicates the relative importance of each input parameter. For example, if for a specific building, $c_2$ is larger than $c_1$, $c_3$, and $c_4$, we would conclude that m′ is the most important parameter. We produced maps of the coefficient values to support examination of the spatial pattern in the strength of the coefficients.

Additionally, we calculated the adjusted coefficient of determination, $R^2$, which indicates the overall ability of a regression to predict h or v using only the model input parameters. A value of the adjusted $R^2$ close to zero indicates that the model input
parameters have little influence on the simulated values of h and v, whereas a value close to one indicates that the model input parameters can perfectly predict the simulated values. Locations with low $R^2$ values indicate areas where the topographic context is more important than model inputs for predicting the value of h or v.





### 4.6 Minimum number of simulations required to produce statistically similar results

In our analysis, we used simulation results generated in a prior study. If a worker wanted to apply the method for forecasting

building damage that we have applied here, they might want to know how few simulations they need to generate similar results. This information may be important in contexts in which time to generate a building damage forecast is limited or if computational resources are sparse. Therefore, as a final analysis, we determined how few simulations would be needed to generate statistically similar results to the best performing forecast that used D-Claw and $hv^2$.

To determine the minimum number of simulations, we calculated how the probability calculated by Equation (1) for each building converged on the value generated using all N simulations as the number of simulations, $N_s$, increased. For each value of $N_s$ considered (5, 10, 15, 20, 25, 30, 40, and 50), we made 30 independent random samples of $N_s$ simulations taken from the full set of simulations and calculated the forecast probability of $D_s=1$ using Equation (1) for each building. For each of the 30 samples, we then determined whether each building would be forecast the same or differently as the case in which all

simulations were used. Finally, we calculated the threat score (Equation (11)) for each bootstrapped sample. We expect that as the number of simulations increase and the bootstrapped samples becomes more statistically similar to the full set of simulation, the threat score will increase. Choosing a minimum number of simulations would require choosing a critical value for the threat score, and here we arbitrarily choose 0.9.

## 5 Results

The results include the logistic regression fit for the fragility function that relates h to $D_s$; 30 forecasts of building damage, generated using three models, h or $hv^2$, and five event magnitude forecast biases; performance assessment of these forecasts, the sensitivity of h and v to model input parameters, and an analysis of how few simulations are needed to generate similar results.

### 5.1 Logistic regression fit

The logistic regression to predict $D_s$ with h (Equation (2) generated estimated values and an assessment of statistical significance for two coefficients, $\beta_0$ and $\beta_1$ (Table S5, Figure S2a). The two fitted coefficients were both significant at the 99% confidence level. To determine the optimal discrimination threshold, we evaluated how the bias and threat score changed as a function of the discrimination threshold (Figure S2b). We found the bias was equal to 1 and the threat score was equal to its maximum at a discrimination threshold of 0.5 (section 4.2.1). Thus, for the purposes of classifying building into $D_s=0$ and

$D_s=1$ we used a discrimination threshold of 0.5. This discrimination threshold corresponds to a depth threshold between $D_s=0$ and $D_s=1$ of 0.47 m.



## 5.2 Forecast performance

We produced maps of forecast building damage and evaluated forecast performance based on the ability of forecasts to represent the observed pattern of building damage. Additionally, aggregated measures of forecast performance supplement
evaluation based on maps and document how forecasts change if the event magnitude forecast bias is larger or smaller than was observed. Taken together, the results indicate that only forecasts made with D-Claw using $hv^2$ can correctly forecast the observed number and spatial pattern of building damage from the Montecito event. We begin the presentation of the results by describing the aggregate measure of forecast performance because they contextualize the spatial patterns with summary statistics like the bias, B, which compares the forecast and observed number of damaged buildings.

### 5.2.1 Aggregate measures of performance

Forecast performance varies among the models and fragility function options (Figure 4 depicts the set of simulations classified as having unbiased event magnitude), with only D-Claw having a bias near one and a threat score of 0.25, comparable to the highest observed for any forecast. Because Figure 4 depicts a Roebber (2009) diagram, a graphical layout with beneficial geometric properties, we describe them before describing the results further. Recall that FAR is the fraction of buildings
forecast as damaged that are not correct, and H is the fraction of buildings observed as damaged that were forecast correctly. Thus, 1–FAR represents the fraction of buildings forecast as damaged that were forecast correctly. A forecast with no true positives would plot at (0,0), and a forecast with no false positives or false negatives would plot at the position (1,1). Forecasts that plot in the upper left half of Figure 4 have more false positives than false negatives, and thus have a bias of greater than one. The converse is true for the lower right half of the diagram. As the proportion of true positives increases relative to false
negatives or false positives, the threat score increases, and a forecast would plot closer to the upper right corner. Forecasts that lie on the same constant value of threat score contour line differ only in the ratio of false positives and false negatives, with more false positives in the upper left and more false negatives in the lower right.

First, we will discuss only forecasts made with an unbiased event magnitude. Later in the results we will discuss how forecast
performance changes with different event sizes. For all models, forecasts that used h have a large, positive bias, whereas the bias for the forecast that use $hv^2$ depends on the model (Figure 4). Forecasts that use h have biases between 2.38 and 3.39, indicating that forecasts that use h predict damage to more than twice as many buildings as was observed (all forecast performance metrics provided in Table S6). The bias of forecasts that used $hv^2$ were low for RAMMS and FLO-2D (0.76 and 0.42, respectively), whereas the bias for the forecast that used $hv^2$ was 1.43 for D-Claw.

The highest threat scores, TS, are associated with the forecasts made with h and the forecast made with $hv^2$ and D-Claw (Figure 4). These four forecasts had threat scores that range between 0.21 and 0.26. In contrast, the two forecasts made with h and either RAMMS or FLO-2D had lower threat scores of 0.17 and 0.13, respectively. Across all six forecast options, most of the



variation in B and TS comes from variation in H rather than variation in 1–FAR. This indicates that across all forecast options
the fraction of buildings forecast as damaged that were not correct stayed constant, whereas the fraction of buildings observed
as damaged that were correct changed.

### 5.2.2 Spatial distribution of forecast building damage

Maps of produced forecasts depict the spatial variation in the probability a building was damaged ($D_s=1$) for each model using
fragility functions based either on h or $hv^2$ (
Figure 5 or Figure 6, respectively). In this section we will discuss only the maps made with the unbiased event magnitude, or
the central column (

Figure 5b,e,h or Figure 6b,e,h). Forecasts made with h and the forecast made with $hv^2$ and D-Claw predict building damage
over the entire portion of the runout path, consistent with observed damage (

Figure 5 and Figure 6). Combining the aggregate measures of performance presented in the previous section with the spatial
pattern presented here indicates that the forecasts made with any model using the fragility function based on h will generate
the correct pattern of building damage but with 2-4x the number of buildings damaged.

The forecast generated using $hv^2$ and D-Claw predicts a high likelihood of building damage in the southern portion of the
alluvial fan, consistent with the observed pattern of building damage (Figure 1) and has a bias of 1.43. In contrast, forecasts
made with RAMMS and FLO-2D using $hv^2$ do not forecast a high likelihood of building damage in the southern, distal portion
of the alluvial fan and concentrate buildings with a high probability of building damage near the runout path apexes. Based on
the combination of aggregate performance and spatial pattern, we conclude that only forecasts made with D-Claw and $hv^2$ can
correctly forecast both the correct number and spatial pattern of buildings.

We generated a map of the location of true positive, false negative, and false positive buildings for the forecasts made with
$hv^2$ because the aggregate performance measures indicated that all models produced forecasts with low threat scores. We used
a 50% probability threshold to classify each building in the forecasts depicted in Figure 6b,e,h into true positive, false positive,
false negative, and true negative (Figure 7 depicts all categories except for true negative). All three models have a similar
pattern of false negatives (buildings that were damaged in the 2018 event but were not forecast as damaged) and true negatives
(unaffected buildings). The biggest difference among the three models is in the pattern of true positives and false positives,
both cases for which buildings were forecast as damaged. RAMMS and FLO-2D have no true positives and false negatives in
the southern, distal portion of the Montecito creek runout path or much of the San Ysidro runout path, whereas D-Claw
correctly forecasts building damage in these areas.



*5.2.3 Impact of the event size on performance*

We can evaluate the role of the event size on forecast performance by examining how the forecast maps and aggregate performance measures change as the event magnitude forecast bias category changes (

Figure 5, Figure 6, Figure 8). This analysis documents how incorrect a building damage forecast might be should the size of the rainstorm have been forecast as larger or smaller than the observed event.

All models and fragility function methods show a similar pattern in performance as the event magnitude forecast bias changes from very overforecast to very underforecast. As event magnitude varies from very overforecast to very underforecast, the forecasts for all models and both h and hv$^2$ trace a path from high H and low 1-FAR to low H (Figure 8). This pattern is consistent with expectations: a high hit rate and large number of false positives when the event magnitude was overforecast (bigger than observed) and a low hit rate when the event magnitude was underforecast (smaller than observed).


Both the spatial pattern and number of buildings damaged is sensitive to the event magnitude forecast bias (

Figure 5 and Figure 6). For the forecasts made with h (

Figure 5), building damage is predicted over most of the runout path extent for all models and all event forecast bias categories, but the extent of forecast damage is wider for the overforecast cases. The results for the forecasts generated with h contrast

with those generated with hv$^2$ in that the latter are more sensitive to both model used and event magnitude forecast bias (Figure 6). RAMMS and FLO-2D do not predict building damage in the distal portions of the fan for any event magnitude forecast bias category. D-Claw predicts a wider area of damage over the entire fan length as the event size increases. These results indicate that matching the correct number and pattern of damaged buildings requires forecasting the event size correctly.

**5.3 Spatial pattern in predicting h and v**

Because D-Claw was the highest performing model based on both the threat score and the bias (Section 6.1), we evaluated two linear regressions at each building to predict h and v with the four model input parameters (Figure 9 and Figure 10; Table S2 indicates how many simulations were used in each regression). The results of this analysis indicate that the event size is the most important model input for predicting both h and v across the alluvial fan. Recall that we standardized both this model input parameters and the model outputs to make the results comparable.


Across the alluvial fan, the event size, represented by the parameter $\log_{10}(V)$, had the largest standardized regression coefficient and was statistically significant (>90%) for predicting both h and v (Figure 9a and Figure 10a). The regression coefficient associated with $\log_{10}(V)$ was always positive, indicating that an increase in event size resulted in an increase in h or v. At most buildings, the regression coefficient for $\log_{10}(V)$ was an order of magnitude larger than the other three parameters.






The amount of cross-simulation variance in h or v explained by the model parameters varied across space, with the highest adjusted $R^2$ values in the central axes of the channelized runout paths (Figure 9e and Figure 10e). Less variance in h or v was explained with distance from the main runout paths, the locations where the event size showed the largest degree of importance.

Of the three remaining parameters, only m´ and $\log_{10}(k´)$ had statistical significance in predicting h or v across the inundated area (Figure 9b,c and Figure 10b,c). A larger value of m´, the difference between the initial and critical solid volume fraction, was associated with an increase in h adjacent to the observed runout paths and a decrease in h in the areas with observed inundation. A larger value of m´ was generally associated with an increase in velocity. A larger value of $\log_{10}(k´)$ was associated with an increase in h in the upper portion of the San Ysidro Creek runout path and a decrease in h elsewhere. Finally,
a larger value of $\log_{10}(k´)$ was associated with lower velocities, except for the upper portion of the San Ysidro Creek runout path.

### 5.4 Number of simulations required

The result of our final analysis indicate that 20-25 D-Claw simulations are needed to generate statistically similar results to those presented with the full set of simulations considered here (Figure S3). The subsampling analysis generated a threat score
measure for each bootstrapped sample that measured how well the forecast based on the bootstrapped sample matched the forecast generated with all simulations. Across all models and all event magnitude forecast bias categories, the threat score increased with increasing number of samples, exceeding 0.90 for D-Claw with 20 simulations (Figure S3c). Obtaining statistically similar results with either RAMMS or FLO-2D would require more simulations than with D-Claw (Figure S3a,b). Both of these models produce lower threat scores for the same number of subsampled simulations. This result indicates that
RAMMS and FLO-2D are both more sensitive to their input parameters than D-Claw.

## 6 Discussion

We discuss the implications of the overall forecast performance, the implications for how debris-flow runout models are evaluated, and methodological limitations.

### 6.1 Forecast performance

The building damage forecast of $D_s$ using $hv^2$ produced by the D-Claw model is the highest performing approach when considering both the number of true positive, false positive, and false negative predictions (Figure 7), but also the spatial pattern of building damage (Figure 6h). The results are consistent with prior work indicating that debris-flow depth alone is not sufficient to forecast building damage (Luo et al., 2023). Only D-Claw forecasts building damage in the southern, distal portions of the Montecito Creek runout path or for much of the San Ysidro runout path. Both locations are places where
buildings were damaged more than 50% (Figure 2) and include the residences of deceased victims. The other two models





produced similar false positive ratios for the same event magnitude forecast bias, but they produced smaller hit rates and associated smaller threat scores (Table S6). RAMMS and FLO-2D forecast building damage near the alluvial fan apex and did not match the observed pattern of building damage. This result implies that RAMMS and FLO-2D do not maintain high peak volume flux values over the portion of the alluvial fan that experienced high volume flux during the Montecito event.


What differences among the three models explain this performance difference? The most notable difference among the three models is that the equations that describe RAMMS and FLO-2D represent flow resistance with a specified relation between shear stress and strain rate, whereas the D-Claw equations allow flow resistance to evolve as pore pressure evolves. Stated another way, debris-flow material movement in RAMMS and FLO-2D always reduces kinetic energy through frictional

dissipation, whereas in D-Claw frictional dissipation of kinetic energy is contingent on the evolving flow dynamics and its strong regulation by coupled pore-pressure evolution. Thus, for a single-phase model like RAMMS or FLO-2D to match the observed inundation extent, the flow must slow prematurely.

Before discussing the spatial patterns of forecast performance more extensively, we discuss the binary classification summary
statistics for the unbiased event magnitude forecast of $D_s$ using $hv^2$ (Figure 6b,e,h). The values for false alarm ratio, hit rate, bias, and threat score indicate that D-Claw has a bias of 1.43, in contrast with the other two models that have biases of 0.42 (FLO-2D) and 0.76 (RAMMS) (Table S6). The models have similar false alarm ratios around 0.65 but differ in their hit rates, with D-Claw having the highest hit rate of 0.48 and the other two models having hit rates of 0.25 and 0.16. These metrics indicate that although all three models generate a similar proportion of false positives to total forecast damaged buildings, D-
Claw forecast as damaged the highest fraction of buildings observed as damaged in the event. Additionally, D-Claw predicted a lower absolute number of buildings that were not forecast as damaged but were observed as damaged (false negatives). These results were unexpected because prior work by Barnhart et al. (2021) demonstrated that all three models produced similar inundation patterns and similar sensitivity to event size. The subsequent analysis of spatial variation provides an explanation for the difference in model performance.
Because of its overall better performance, for the remainder of this subsection, we limit our discussion to the performance of only one forecast method: using D-Claw to predict $D_s$ with $hv^2$. Later in the discussion, we return to intermodel comparison.

### 6.1.1 Spatial pattern of forecast performance

Examination of the spatial pattern of false positives and false negatives in the best performing forecast made with $hv^2$ and D-Claw indicates coherent patches of forecast error that have implications for the reliability of building damage forecasts made
with runout models (Figure 11). We investigated the detailed spatial pattern in this forecast because the threat score was 0.25 while the bias was 1.43, indicating that the number of forecast buildings were correct, but that many forecasts were false positives or false negatives. Examination of the lower portion of the Montecito Creek runout path and the entirety of the San Ysidro Creek runout path indicates coherent patches of false positives and false negatives (regions indicated on Figure 11). In



region I, false positives are clustered around the edge of the flow, and false negatives are intermingled with true positives. In region II, false positives are located on the eastern flow edge, and false negatives are located on the western edge of the flow edge. In region III, flow in a distributary channel to the west of San Ysidro Creek predicts extensive building damage where little was observed, yielding a patch of false positives with few false negatives or true positives nearby. In region IV, many false negatives intermingle with true positives. Finally, in region V, false positives are located to the west of true positives, and false negatives are located to the east of true positives.


The overall threat score value and the spatial correlation of false positive and false negative do not support interpreting the results as reliable at the individual building level even though the building damage forecast is made at the individual building level. Instead, they support interpreting the overall spatial pattern of forecast building damage and assuming that only half of the buildings forecast as damaged are correct, with the remaining half being false positives. Additionally, a similar portion of

buildings classified as undamaged are likely to be damaged and thus false negatives. Additionally, the location of false positives and false negatives is not random, but spatially correlated. The most substantial implication of this observation is that only the broad spatial pattern and number of buildings damaged can be considered as reliable. Later in the discussion, we discuss the implications of this spatial correlation for improvement of debris-flow runout models.

### 6.1.2 Influence of event size

The large variation in event size (means and ranges listed in Table S2) indicates that the spatial pattern of forecast damage is sensitive to the event size as it changes from underforecast to overforecast and the total volume of mobile material increases four-fold. In addition to the total number of buildings forecast as damaged increasing (reflected in the bias values in Table S6), the width of the forecast damage area increases for the forecasts made with h (

Figure 5) and $hv^2$ (Figure 6). This sensitivity to event size is similar to the prior evaluation of the inundated area (Barnhart et
al., 2021) that showed a strong sensitivity to event size that was comparable between the three models.

Taken together, the results from this study and Barnhart et al. (2021) indicate that $hv^2$ is likely a more reliable metric than h for identifying the area impacted by postfire debris-flow runout but that the quality of the forecast depends on how well the event size may be ascertained in advance. Ultimately, how useable maps forecasting $hv^2$ or building damage and what level of
confidence is tolerable is a question for land and emergency management decision makers.

### 6.2 Implications for evaluation of debris-flow runout models

In this section, we first discuss lessons regarding how and with what data to evaluate runout models before turning to implications for improvement.



### 6.2.1 How should debris-flow models be judged?

Because accurately predicting building damage requires forecasting both h and v, building damage is a stricter test of model fidelity than simply matching runout extent or spatially distributed observations of depth. More specifically, a model-data comparison that uses two aspects of the phenomena of interest evaluates the generality of the model, a term borrowed from scholars in philosophy of science who study the practice of modeling (Weisberg, 2013). The generality of a model is an advantageous characteristic for the type of application considered here: use of runout models in locations where few

observational data are available to calibrate model parameters.

The reader may recognize the title of this subsection as referring to Iverson (2003) "How should mathematical models of geomorphic processes be judged?" Indeed, this subsection was influenced by the volume "Prediction in Geomorphology" (Wilcock and Iverson, 2003), most notably the contributions by Iverson (2003) and that of Furbish (2003). Iverson (2003)

discussed a hierarchy of data for model tests, arguing that experiments, with known initial and boundary conditions, and independently constrained values for model parameters provide the most stringent tests for the evaluation of any model of a physical system. But what does Iverson (2003) mean by stringent? And what is the purpose of evaluating models? For insight into these questions, we rely on ideas about different approaches modelers may take and fidelity criteria modelers may use in evaluating models that were put forward by Michael Weisberg in his book "Simulation and Similarity" (Weisberg, 2013).


Weisberg (2013) identifies multiple ways scientists (1) idealize phenomena of interest to generate models and (2) judge the application of models to observations of specific aspects of phenomena. Here we only describe the approach taken in this work and implicit in the prior work of Iverson (2003) and that of Furbish (2003). In this, and similar work, we are concerned with the practice of a scientist comparing models with data when the purpose of modeling has multiple aims. A scientist may want

to know what set of equations best describes the complex phenomena of debris flow runout, in part for the purpose of understanding the physical world and in part because having such a set of equations is of practical use for forecasting a hazard. Finally, the scientist likely does not expect any set of equations to be completely correct because of the complexity of the phenomena. Because the scientist has many aims, they determine that a model that can predict more aspects of the phenomena (e.g., location, depth, speed) is better than one that can predict fewer. Weisberg (2013) would describe this approach as one

that values generality as the fidelity criteria for determining that one model is better than the other. It naturally follows that to test whether a model scores better based on generality one would need observations of more than one aspect of the phenomena of interest. On this topic, we would be remiss if we did not acknowledge the difficulties in making direct observations of debris flows outside of controlled experimental settings. The most common observation is typically the maximum extent of impacted area, a single aspect of the phenomena of debris-flow runout, and one that is highly spatially correlated. Sometimes

observations of debris-flow deposits constrain the total volume of the event. Mudlines may accurately record or overestimate peak flow depths. Superelevation of flow around bends and upstream-downstream pairs of mudlines on the same object can be used to infer flow velocity. Long-period seismic records can record the acceleration and deceleration of the center of mass.



Finally, as we show here, the buildings damaged in the wake of a debris flow reflect more than the peak depth. Notably, except for the long-period seismic records, none of these observations are time-variable. Instead, they represent a maximum or critical value for at an individual location.

A synthesis of this contribution and the prior contribution of Barnhart et al. (2021) provides a concrete example of using generality to evaluate three models because a comparison can be made between model evaluation based solely on one target, debris-flow depth, and two targets, based on both depth and velocity. Although the Montecito event was not a laboratory experiment, with known initial and boundary conditions and constrained parameter values, the exception quantity of observational data collected as part of the response effort and by subsequent authors (Oakley et al., 2018; Kean et al., 2019b; Lukashov et al., 2019; Lancaster et al., 2021; Alessio et al., 2021; Morell et al., 2021) make it akin to a natural experiment (Tucker, 2009). In Barnhart et al. (2021), the authors demonstrated that these same three considered models performed similarly well at the prediction of debris-flow extent and depth. This finding contrasts with the more stringent model test implemented here, in which predicting building damage is a proxy for predicting volume flux, or both h and v. In this second test, only D-Claw performed well at forecasting the spatial patterns of building damage, which may indicate it performs well at forecasting peak volume flux.

In summary, models that can match observed patterns of building damage demonstrate better generality than those that can just match observed runout extent because forecasting building damage requires both depth and velocity. D-Claw is a more general model than RAMMS or FLO-2D because the way its equations handle flow resistance allows it to represent both depth and velocity better than either of the considered alternatives. Studies that interrogate the evaluative capacity of different characteristics of debris-flow runout (e.g., the evaluation of adverse slopes by Iverson et al., 2016) may provide direction for which targets and where in the landscape debris-flow runout models may be most effectively tested.

*6.2.2 Spatial correlation in forecast error*

Finally, the spatial patterns of false positives and false negatives in Figure 11 point to potential improvements in the D-Claw model physics. Field observations presented in Kean et al. (2019b) indicate that 1–2 m boulders were dropped at the top of the distributary channel within region III but that few boulders made it farther down the distributary channel. The influence of this deposition results in the difference in damage experienced by the buildings in Figure 2b and Figure 2c. The former was closer to the fan apex, but along the distributary channel it was impacted by 0.7 m of mud. It contrasts with the latter, which was inundated nearly to the eaves by boulders. The role of deposition, including different grain size classes, influences the nature of building damage and indicates that improving the capacity for D-Claw to simulate the deposition of material may improve the spatial pattern of volume flux across the landscape. The concentration of false positives along the east bank of Montecito Creek in region I indicates that flow may not have been sufficiently confined as it moved downstream. This may indicate that improved representation of channelizing processes, such as levee development or channel scour, may improve building damage





forecasts (Jones et al., 2023). Finally, the systematic striping of false positive, true positive, and false negatives in regions II and V in Figure 11 indicate that the simulated dominant flow was not going in the correct direction. In the case of region I, simulated flow was trending to the southeast whereas in the event it trended to the south, and in the case of region V simulated flow trended to the southwest whereas in the event it trended to the south. Both regions are areas where flow was not confined

by the topography, and these systematic errors may indicate that additional of mechanisms for self-channelization are important model improvements.

### 6.2.3 Source of building-level variance

Across the landscape, the most important parameter, by an order of magnitude, for predicting h and v is the event size $\log_{10}(V)$ (section 5.3). This result is expected because without debris-flow material, the area will not be inundated. The importance of

event size has long been recognized and is the basis for the success of empirical scaling relations relating event size and impacted area (Iverson et al., 1998). Because event volume is the most important model input for h and v, efforts to reduce uncertainty in debris-flow runout and building damage forecasts would be best served by reducing uncertainty in event size. Such efforts likely include process-based and empirical approaches focused on sediment recruitment from hillslopes and channels.


We can isolate the role of topography in controlling areas that may be impacted by runout by evaluating the strength of the regression predicting the simulated values of h and v using model input parameters as the independent variables (section 5.3, Figure 9, Figure 10). The portions of landscape where the regression has a low value for the adjusted $R^2$ (Figure 9e, Figure 10e) indicate areas where the topography is just as, if not more, important than event size and mobility for influencing whether

the area will be impacted by runout. As should be expected, the values of h and v are more predictable by the four model inputs in areas of channelized flow, such as in the upper reaches of Montecito, San Ysidro, and Romero Creeks, as well as the distributary channel that branches to the west from San Ysidro Creek.

Both m´ and $\log_{10}(k´)$ are important to predict h and v, but with different spatial patterns and both to a lesser degree than the

event size. Larger values of m´ mean that the initial specification of the debris-flow material has a higher solid volume fraction, closer to the critical solid volume fraction. All values of m´ used in Barnhart et al. (2021) were negative, meaning that initial motion increased pore pressure and decreased intergranular friction. Higher values of m´ produced a complex pattern in flow thickness and faster peak flow across the landscape. Prior work investigating the sensitivity of runout dynamics to m´ in the context of the 2014 State Route 530 landslide near Oso, WA, indicated that smaller, more negative values of m´ were

consistently associated with faster runout and larger values of total momentum (Iverson and George, 2016). Our results are not conclusively in conflict with the prior results because larger values of m´ may be associated with larger longitudinal stress gradients. Numerical experiments with a simpler geometry may illuminate an explanation for these patterns.





In contrast with m′, the spatial pattern of the importance of $\log_{10}(k′)$ is more straightforward to understand. Interpretation is
aided by recalling that $\log_{10}(k′)$ is the ratio of the two timescales that govern downslope motion and pore pressure diffusion
such that as pore pressures decrease, intergranular friction increases. It is expected that smaller values of $\log_{10}(k′)$, reflecting
a longer timescale of pore pressure diffusion and a longer duration of elevated pore pressures, are associated with faster flow
over most of the impacted area.

## 6.3 Limitations and implications for hazard assessment

Our results indicate that D-Claw combined with the FEMA Hazus tsunami fragility function method can forecast the spatial
patterns of building damage better than either RAMMS or FLO-2D. We conclude the discussion by describing data limitations
and implications for applying the presented methodology in other locations.

### 6.3.1 Building and topography data

A notable area for improvement is in the dataset of building characteristics used. In this study we used building geometry from
Open Street Map, chosen for its ease of use. Although these building footprints afford estimation of the area, width, and
location of individual buildings, they do not provide information about the construction material or building age. Application
of the Hazus fragility functions used here to predict damage class with $hv^2$ and more detailed information about construction
material may improve the overall quality of the predictions. Use of this type of information may also make the approach more
reliable in areas where the dominant type of building is not a light wood-frame residential building. Notably, unlike the
empirical fragility function we fit relating debris-flow depth to damage (section 5.1), the Hazus fragility function method is
not specific to wood-frame residential buildings so it may be applicable to other building types using the appropriate strength
parameters.

An additional area for potential improvement is in the representation of the buildings within the runout model simulations. In
this study, we did not directly represent the buildings but instead used a 5-m bare-earth digital elevation model. Our approach
assumed that the details of debris flow-building interaction over at the spatial scale of the entire runout path and at a simulation
resolution of 5-m was not necessary to represent the pattern of observed building damage. Additional research that evaluates
how forecast performance changes with smaller computational grid cells or digital elevation models that include the buildings
may indicate the validity of this assumption.

### 700 6.3.2 Depiction of hazard forecasts

For the observed event size, the best combined forecasts developed here have a hit rate of around 50% and a threat score of
25% (Table S6). Furthermore, the spatial patterns of forecast building damages included spatially correlated patches of false
positives and false negatives (Figure 11). Consequently, the forecast performance does not support the conclusion that a similar



forecast for another event could be reliably interpreted at an individual building level. This motivates the question: what ways of depicting a building damage forecast reflect the inherent uncertainty in building-level predictions? One option might be to smooth the prediction, using a spatial scale of smoothing that reflects the length-scale of systematic forecast error depicted in Figure 11. Such an approach would inherently overestimate the number of buildings damaged but might be more reliable at capturing the areas with true negatives. An alternative depiction might focus on the number of buildings damaged along the major runout paths and present the forecasts along the runout paths rather than in plan view. A challenge with this type of depiction might be that the runout paths are inherently dependent on the model simulations and may be complex in plan view. If required information about building geometry and material type are not known, a representative building might be used. Additionally, many other factors beyond a detailed assessment of damage potential may be relevant for generating a debris-flow inundation hazard. A multi-stage hazard assessment that describes areas susceptible to inundation and the smaller area susceptible to damage may provide an approach to depiction that errs on the side of caution. Which, if any of these depiction options would be most useful to land and emergency management personnel is itself another research question that could be assessed with the methods of user needs assessment and user-centered design.

### 6.3.3 Computational requirements

The finding that $hv^2$ produced by D-Claw provides a reliable forecast of the number and spatial distribution of damaged buildings prompted evaluation of the minimum number of simulations needed to generate a statistically similar forecast. D-Claw has substantially larger computational requirements than the other two models presented here (600-900 core hours, as compared with two core hours for the other two models, Barnhart et al., 2021). The bootstrapping analysis documented how variability in forecasts decreases with an increased number of sampled simulations and that 20 simulations is sufficient to reproduce the results provided in this contribution (Figure S3). Even with a smaller number of simulations, it may be computationally intractable to apply the methodology described here across large areas (tens to thousands of square kilometers). Evaluation of the sensitivity of building damage forecasts to computational grid size may result in computationally tractable approaches suitable for large areas. Furthermore, understanding the relative usability of forecasts made with faster models and known large bias relative to slower models with lower bias may guide development of the most usable hazard assessment methods. One possible approach may be to run faster, less accurate models over the entire fire-impacted area to generate an assessment of the areas susceptible to impact by debris-flow runout, and then only run D-Claw simulations in areas where the faster, less accurate models indicate that debris-flow material may interact with residences, roads, and other infrastructure of interest.

## 7 Conclusions

The objective of this study was to evaluate the capacity of runout models, in combination with fragility functions, to forecast building damage due to postfire debris flows. We documented the relative performance of three runout models using two

fragility function methods. We found that forecasts based on depth have a higher bias than forecasts based on volume flux, forecasts are sensitive to the event size, and that only D-Claw correctly forecasts the observed pattern of building damage and number of buildings damaged. Despite having the best performance—a hit rate of about 1/2, a false alarm ratio of 2/3, and spatially correlated false positives and false negatives—these results do not support interpreting forecasts of damage using D-Claw generated $hv^2$ at the individual building level.


The implications of this work are practical and fundamental. The practical implication is that probabilistic forecasts of damage for wood-framed buildings can be made with the D-Claw model combined with the Hazus model without detailed back calculation of parameters. Notably, the Hazus approach is not unique to wood-frame buildings such that a similar method may work well with other building types. The fundamental implication is that spatially variable building damage contains more

information about debris-flow dynamics and is a better test of model generality than the extent of impacted area. Intermodel comparison indicates that D-Claw can reproduce the spatially variable patterns of h and v (and therefore $hv^2$) needed to reliably forecast building damage. Unlike RAMMS and FLO-2D, D-Claw can reproduce flows that move rapidly over the landscape without substantial dissipation of energy because high pore pressures result in low intergranular friction.

Finally, examination of the spatially correlated location of forecast errors and the sensitivity $hv^2$ to D-Claw input parameters points to targets for improvement. First, the spatially correlated errors are consistent with patterns of deposition and self-channelization observed during the event. Second, the dominance of event size in influencing the simulated pattern of peak $hv^2$ further emphasizes the importance of constraining the mechanisms that influence mobilized debris-flow volume, including entrainment of sediment on hillslopes and scour in channels, and understanding how the rate of sediment mobilization depends

on rainfall intensity.

## Data availability

Data used in this study were downloaded from Open Street Map (OpenStreetMap contributors, 2021) or are provided in Kean et al. (2019a). Model output from Barnhart et al. (2021) is provided in Barnhart (2023).

## Author contributions

KB conceived of and implemented the study based on conversations with JK. CM provided guidance on statistical model application and evaluation. FR and JK provided input on interpreting the results based on their observations of the Montecito event. KB wrote the original manuscript, and all authors contributed to its revision.

## Competing interests

The authors declare that they have no conflict of interest.





**Acknowledgements**

Any use of trade, firm, or product names is for descriptive purposes only and does not imply endorsement by the U.S. Government. Comments from and discussion with Dick Iverson, Dave George, and Dave Hyman improved the design and interpretation of this study. Reviews from Phil LeSeur, Jacob Woodard, Rex Baum, Brian Shiro, and Janet Carter improved the content and clarity of the manuscript.

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



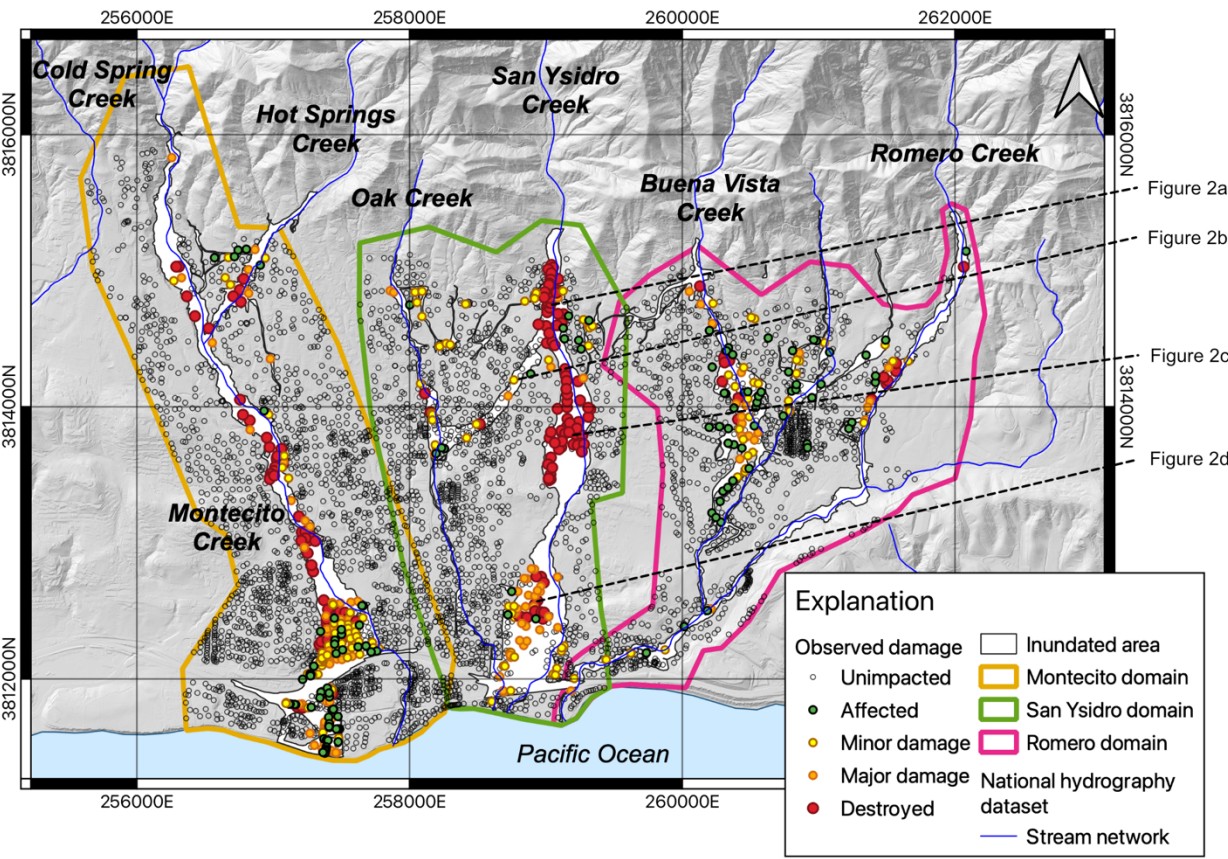

**Figure 1.** Map depicting the location of all five building damage classes considered in this study and the names of creeks in the study area. The yellow, green, and pink lines in each panel depict the extent of the three simulation domains, Montecito, San Ysidro, and Romero, respectively. The white region depicts the mapped extent of debris-flow inundation. Undamaged buildings not within one of the three simulation domains are not shown. The dashed lines indicate the locations of building damage examples (Figure 2). Coordinates in this and following maps are easting and northing in Universal Transverse Mercator zone 11N. Basemap and hydrography dataset from U.S. Geological Survey (2017, 2022).





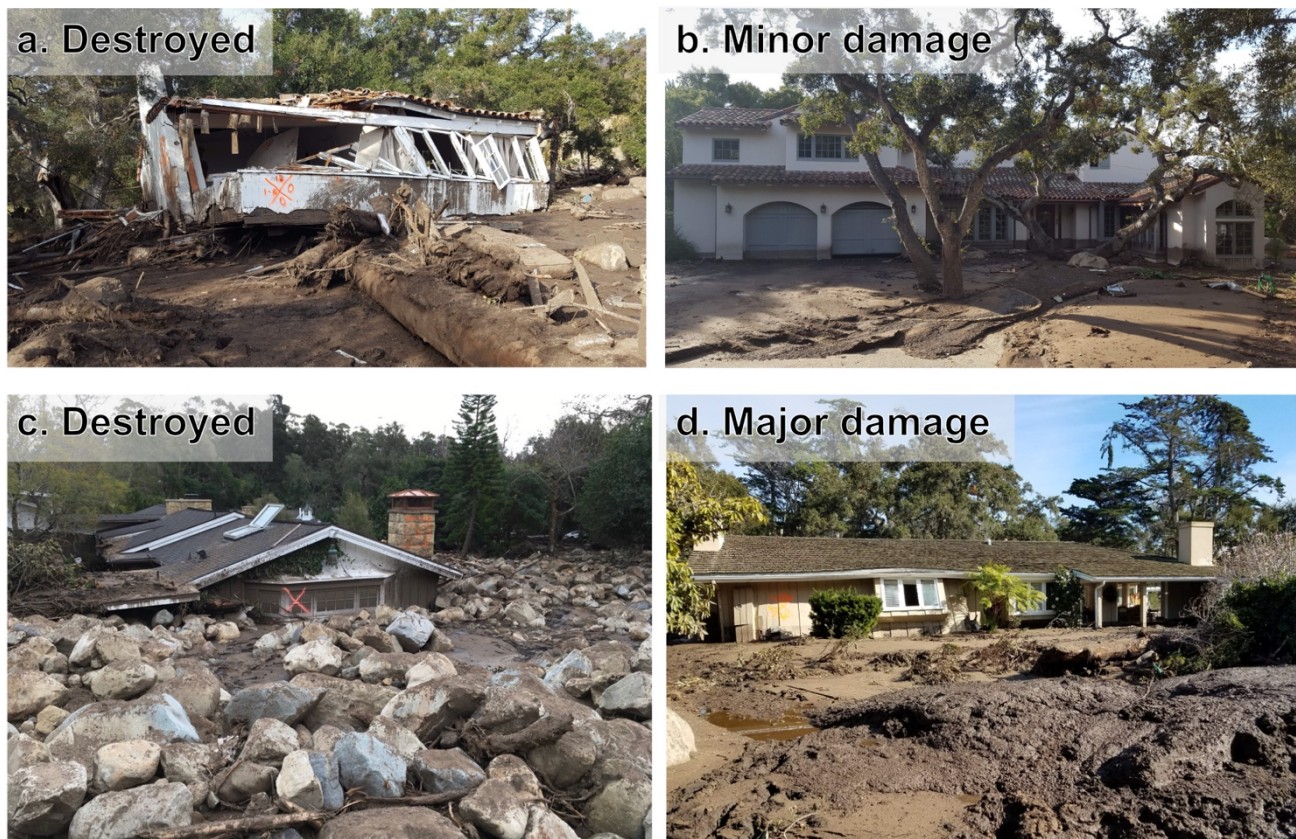


**Figure 2. (a,c,d) Examples of buildings that were classified as sustaining major damage or destroyed from different locations in the San Ysidro runout path (locations depicted in Figure 11). Some buildings had the entire first floor wiped out (a, CAL FIRE building damage ID 437), others were inundated by boulders (c, CAL FIRE ID 268), and others were impacted by mud (d, CAL FIRE ID 100). Panel b (CAL FIRE ID 389) depicts a building that had minor damage. All photos from Kean et al. (2019a).**






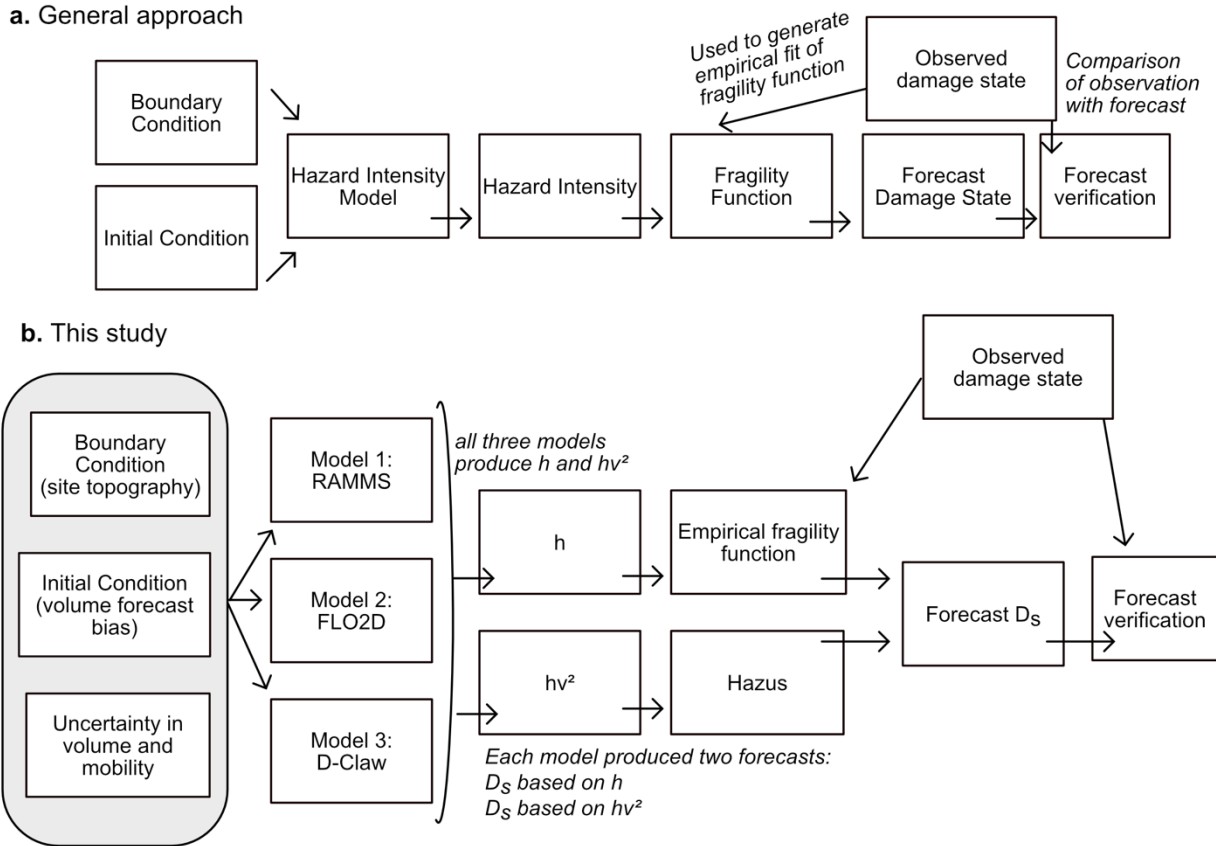

**Figure 3. Conceptual diagram describing (a) a general approach to using a model to generate a forecast of building damage states and (b) the approach used in this study to predict the simplified damage state ($D_s$) with either maximum debris-flow depth (h) or volume flux ($hv^2$). Observed building data were used for two purposes: alongside observations of debris-flow depth the data were used to generate of a fragility function, and alongside model predictions the data were used to evaluate forecast results.**



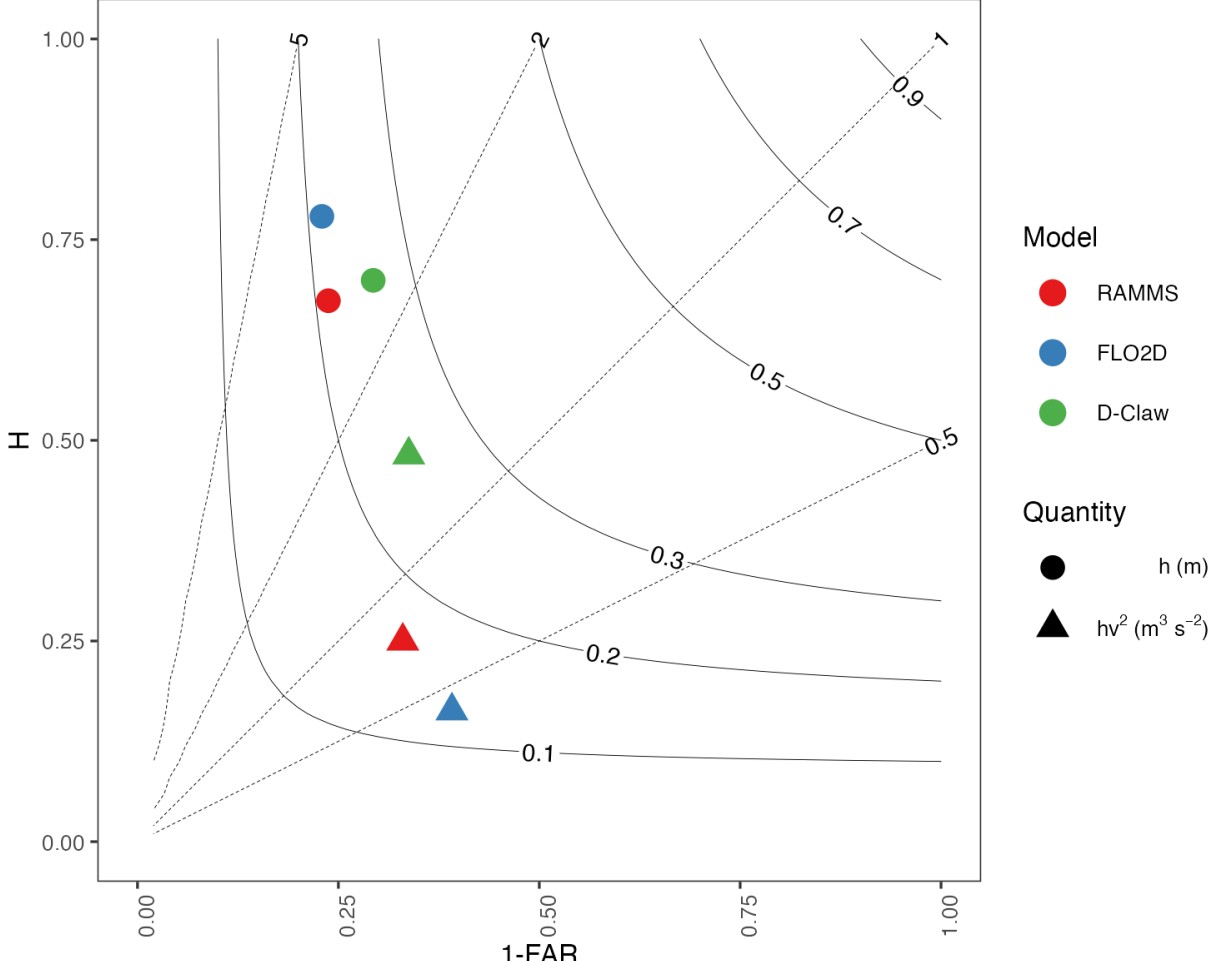

**Figure 4. Roebber (2009) performance diagram comparing the performance of the four candidate forecast options for three models and three simulation domains. Based on the number of true positives (TP), false positives (FP), and false negatives (FN), the hit rate (H), false alarm ratio (FAR), bias (B), and threat score (TS) are defined as follows: H=TP/(FN+TP), FP/(TP+FP), B=(TP+FP)/(TP+FN), and TS= TP/(TP+FN+FP). Each dot represents a forecast with an unbiased event magnitude. The thin solid black lines depict contours of the threat score, and the thin dashed lines depict contours of the bias. Perfect performance is found in the upper right corner. Forecasts generated with $hv^2$ typically have a bias closer to 1 than those generated with h.**



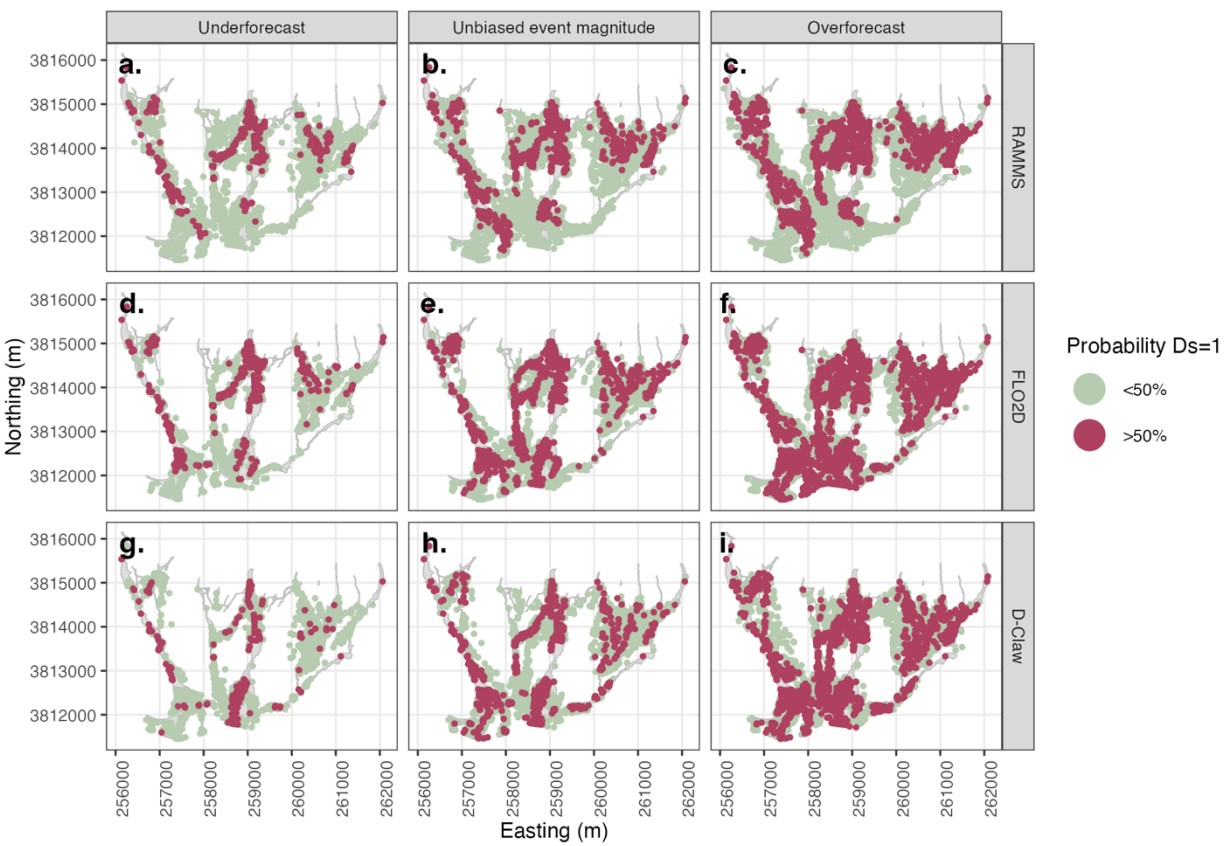

**Figure 5. Example forecast map using maximum debris-flow depth (h) to predict simplified damage state ($D_s$). Rows depict model used, and columns depict three considered levels of event magnitude forecast bias. Each dot represents an individual building with the color depicting the probability of the damage state exceeding zero. All models predict similar patterns of damage, and for the unbiased event magnitude, all predict more buildings are damaged than were observed as damaged in the 2018 Montecito event.**



**Figure 6. Example forecast map using the maximum volume flux (hv²) to predict simplified damage state (Ds). Rows depict model used, and columns depict three considered levels of event magnitude forecast bias. Each dot represents an individual building with the color depicting the probability of the damage state exceeding zero. For the unbiased event magnitude, only D-Claw predicts the pattern of damage observed in the 2018 Montecito event.**





**Figure 7. Classification of the forecast map using the maximum volume flux ($hv^2$) to predict simplified damage state ($D_s$) into true positive (TP), false negative (FN), and false positive (FP) for each building. True negatives are not depicted and are consistent across all models. Rows depict model used. Each dot represents an individual building. Models all produce similar patterns for FN. RAMMS and FLO-2D produce similar patterns for TP and FP, and do not forecast building damage in the southern, distal portion of the runout zone. D-Claw successfully forecasts building damage in the distal portion of the runout zone.**





925

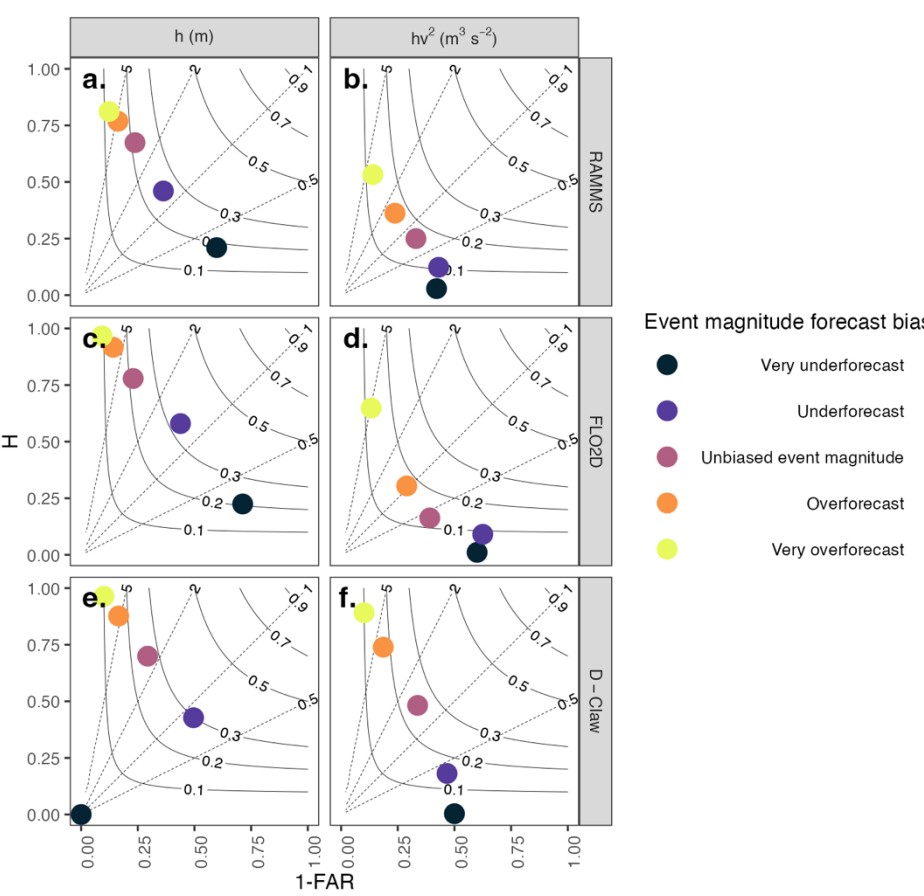

**Figure 8. Roebber (2009) performance diagram comparing the performance of forecasts as the event magnitude varies from very underforecast (black) to very overforecast (yellow). Rows and columns depict model and domain, respectively. Each dot represents a forecast. The thin solid black lines depict contours of the threat score, and the thin dashed lines depict contours of the bias. Perfect performance is found in the upper right corner of each panel. The correct event magnitude (labelled "Unbiased") is typically associated with the highest threat scores, emphasizing the importance of forecasting the event size.**

930





**Figure 9. Regression coefficients (a-d) and adjusted R² value (e) for a regression predicting maximum debris-flow depth (h) with model input parameters at each building for the simulations done with D-Claw. The four model input parameters are the event size, log(V); the difference between the initial solid volume fraction and the critical solid volume fraction, m´; the ratio of the timescale of downslope debris motion and the relaxation of pore pressure, log(k´); and the basal friction angle, $\phi_{bed}$. A building is depicted only if the statistical significance of the coefficient (a-d) or regression (e) exceeds 90%. The event size, represented by log(V), explains the most variation in h across the runout path.**





Figure 10. Regression coefficients (a-d) and adjusted $R^2$ value (e) for a regression predicting velocity (v) with model input parameters at each building for the simulations done with D-Claw. The four model input parameters are the event size, log(V); the difference between the initial solid volume fraction and the critical solid volume fraction, m′; the ratio of the timescale of





downslope debris motion and the relaxation of pore pressure, log(k´); and the basal friction angle, $\phi_{bed}$. A building is depicted only if the statistical significance of the coefficient (a-d) or regression (e) exceeds 90%. Similar to Figure 9, the event sizeexplains the most variation in v across the runout paths.

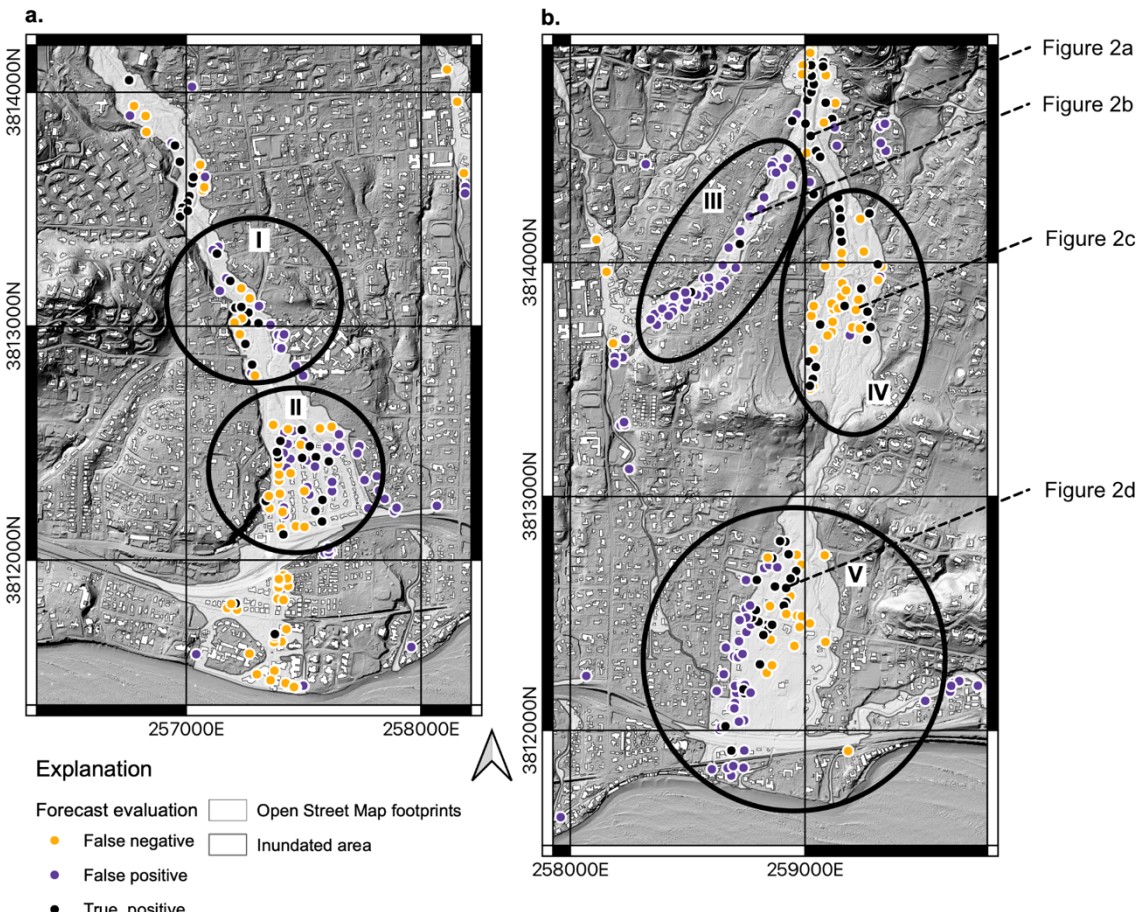

**Figure 11. Maps of the southern part of the Montecito Creek runout path (a) and the San Ysidro runout path (b) depicting the location of true positive, false positive and false negative building damaged forecasts from the forecast of the simplified damage state ($D_s$) with the volume flux ($hv^2$) produced by D-Claw. Black ellipses labelled I-V indicate regions of systematic error in the forecasts discussed in the text. Dashed lines indicate the location of buildings depicted in Figure 2. Basemap from U.S. Geological Survey (2017).**





## Appendix A. Notation

$A_U$, fraction of gravitational acceleration at pushover, unitless

$A_Y$, fraction of gravitational acceleration at yield, unitless

$\alpha_1$, modal mass parameter, unitless

$B$, bias, unitless

$B_w$, the width of the building perpendicular to the flow direction, m

$\beta_0$, $\beta_1$, estimated coefficients in the fragility function analysis, units vary

$\beta_j$, is the lognormal standard deviation associated with damage class $D_s=1$, unitless

$c_0$, $c_1$, $c_2$, $c_3$, and $c_4$, estimated regression coefficients from the regression analysis, units vary

$C_D$, drag coefficient, unitless

$D_s$, simplified damage state, a categorical variable in which 0 is no damage and 1 is damaged, unitless

$F_C$, critical force per unit area, kg m s$^{-2}$

$F_{DF}$, debris-flow impact force per unit area, kg m s$^{-2}$

$F_U$, pushover force per unit area, kg m s$^{-2}$

$F_Y$, yield force per unit area, kg m s$^{-2}$

FAR, false alarm ratio, unitless

FN, false negative, unitless

FP, false positive, unitless

$\phi_{bed}$, basal friction angle, degrees

$\Phi(\cdot)$, cumulative standard normal distribution function, unitless

$h$, peak debris flow depth, m

$hv^2$, peak debris flow volume flux, m$^3$ s$^{-2}$

$H$, hit rate, unitless

$\overline{hv^2}$ is the median volume flux, m$^3$ s$^{-2}$

$K_D$, a factor that accounts for uncertainty in loading, unitless

$\log_{10}(k')$, base-10 logarithm of the ratio of the timescale of downslope debris motion and the relaxation of pore pressure, unitless

$\log_{10}(V)$, base-10 logarithm of the total event volume, unitless

$m'$, difference between the initial solid volume fraction and the critical solid volume fraction, unitless

$N$, the number of simulations combined to generate a building damage forecast, unitless

$N_s$, the number of simulations subsampled in the bootstrapping analysis, unitless

$\rho$, the density of the flow, kg m$^{-3}$

$\sigma_{k'}$, standard deviation of $\log_{10}(k')$, unitless



$\sigma_\phi$, standard deviation of $\phi_{bed}$, degrees

$\sigma_{m'}$, standard deviation of m′, unitless

$\sigma_y$, standard deviation of y, units vary

$\sigma_V$, standard deviation of $\log_{10}(V)$, unitless

TN, true negative, unitless

TP, true positive, unitless

TS, threat score, unitless

v, peak debris flow velocity, m s$^{-1}$

W, total building seismic design weight per unit area, kg m s$^{-2}$

$x_b$, a unique identifier for each building, unitless

y, output of interest, h or v, in the regression analysis, units vary

$\zeta$, median value of the volume flux associated with damage class $D_s=1$, m$^3$ s$^{-2}$