# Peer review of "Evaluation of debris-flow building damage forecasts"

_EGUsphere, 2023_

## Referee Comment (RC1)

**Review of the manuscript No. egusphere-2023-1892 '*Evaluation of debris-flow building damage forecasts*' submitted to *NHESS*.**
* * *
Recommendation: ACCEPT

Focus of the paper is on reliable forecasts of building damage due to debris flows.

Relevance: The presented study is the original primary research within the scope of the journal. In this study, the authors evaluated the performance of building damage forecasts for the 9 January 2018 Montecito post-fire debris-flow runout event, in which over 500 buildings were damaged.

Abstract is well written and clearly describes the undertaken study.

Structure: The article is well organized with structured sections.

Introduction presents a background, defines research goals and provides a clear statement of research problem. It presents the purpose of the research investigation which is supported by the pertinent literature. Literature is well referenced and relevant.

Research questions and goal are identified: Application of debris-flow runout models to generate forecasts requires combining hazard intensity predictions with fragility functions that link hazard intensity with building damage.

Motivation is explained: reliable forecasts of building damage due to debris flows may provide situational awareness and guide land and emergency management decisions.

Methods: Methods described with sufficient information: The authors constructed forecasts using either peak debris-flow depth or volume flux as the hazard intensity measure and applied each approach using three debris-flow runout models (RAMMS, FLO-2D, and D-Claw).

Results are reported: The authors generated forecasts which were based on combining multiple simulations that sampled a range of debris-flow volume and mobility, reflecting typical sources and magnitude of pre-event uncertainty.

Discussion interpreted the major outcomes of this study: The authors found that only forecasts made with volume flux and the D-Claw model could correctly forecast the observed number of damaged buildings and spatial patterns of building damage. The authors also found found that both depth and velocity are needed to forecast building damage, comparing debris flow models against spatially distributed observations of building damage is a more stringent test for model fidelity than comparison against the extent of debris-flow runout.

Conclusion The authors concluded that the event size strongly influences the number of damaged buildings and the spatial pattern of debris-flow depth and velocity. Conclusions are well stated, linked to original research question, limited to supporting results and summarized the study with interpretation of facts.

Recommendations for future work: The authors noted that future research on the link between precipitation and the volume of sediment mobilized may have the greatest effect on reducing uncertainty in building damage forecasts.

Actuality, novelty and importance of the research: the authors indicated that forecasts made at the building level reliably reflect the spatial pattern of damage, but do not support interpretation at the individual building level.

Academic contribution: The paper increases the knowledge in predictive forecasting of building damage to debris flow. Thus, The authors remarked that the best forecast only predicted 50% of the observed damaged buildings correctly and had coherent spatial patterns of incorrectly forecast building damage (i.e., false positives and false negatives).

Figures Figures are of acceptable quality, easy to read, relevant and suitable.

Recommendation: This manuscript can be ACCEPTED based on the detailed report above.

With kind regards,

- Reviewer.

24.01.2024.

---

## Referee Comment (RC2)

**Review of the manuscript No. egusphere-2023-1892 '*Evaluation of debris-flow building damage forecasts*' submitted to *NHESS*.**
* * *
Recommendation: accept

Focus of the paper is on reliable forecasts of building damage which are caused by debris flows. Such prognosis may provide situational awareness and guide land and emergency 10 management decisions.

Relevance: The presented study is the original primary research within the scope of the journal. The manuscript meets general criteria of the significance in risk assessment. The study has been conducted in accordance to the technical standards in modelling and spatial data analysis.

Abstract is well written and clearly describes the undertaken study.

Structure: The article is well organized with structured sections.

Introduction presents a background, defines research goals and provides a clear statement of research problem. It presents the purpose of the research investigation which is supported by the pertinent literature. Literature is well referenced and relevant.

Research questions and goal are identified: Developing methods of debris-flow runout modelling to generate forecasts which requires combining hazard intensity predictions with fragility functions that link hazard intensity with building damage.

English language: acceptable. Clear, unambiguous, professional English language used throughout.

Data used in this study are described: The authors evaluated the performance of building damage forecasts for the 9 January 2018 Montecito postfire debris-flow runout event, in which over 500 buildings were damaged.

Methods: Methods described with sufficient information: The authors constructed forecasts using either peak debris-flow depth or volume flux as the hazard intensity measure and applied each approach using three debris-flow runout models (RAMMS, FLO-2D, and D-Claw). The workflow is well structured. Generated forecasts were based on combining multiple simulations that sampled a range of debris-flow volume and mobility, reflecting typical sources and magnitude of pre-event uncertainty.

Results are reported: The authors found that only forecasts made with volume flux and the D-Claw model could correctly forecast the observed number of damaged buildings and the spatial patterns of building damage. They also noted that the best forecast only predicted 50% of the observed damaged buildings correctly and had coherent spatial patterns of incorrectly forecast building damage (i.e., false positives and false negatives).

Discussion interpreted the major outcomes of this study: The results obtained by the authors indicate that forecasts 20 made at the building level reliably reflect the spatial pattern of damage, but do not support interpretation at the individual building level. The authors found the event size strongly influences the number of damaged buildings and the spatial pattern of debris-flow depth and velocity.

Conclusion The authors concluded that future research on the link between precipitation and the volume of sediment mobilized may have the greatest effect on reducing uncertainty in building damage forecasts.

Actuality, novelty and importance of the research: The authors found that both depth and velocity are needed to forecast building damage, comparing debris flow models against spatially distributed observations of building damage is a more stringent test for model fidelity than comparison against the extent of debris-flow runout.

Academic contribution: The paper increases the knowledge in methods of risk assessment and prognosis of potential consequences of geological hazards.

Figures Figures are of acceptable quality, easy to read, relevant and suitable.

Recommendation: This manuscript can be accepted based on the detailed report above.

With kind regards,

05.02.2024.

---

## Author Comment (AC1)

*Response to reviewer comments: egusphere-2023-1892*

Title: Evaluation of debris-flow building damage forecasts
Authors: Katherine R. Barnhart, Christopher R. Miller, Francis K. Rengers, and Jason W. Kean

Here we summarize changes made to the manuscript based on the comments from two reviewers. We thank both reviewers for undertaking reviews of the manuscript.

**Reviewer Comments 1 and 2 (Referee #1, Polina Lemenkova)**

Referee #1 submitted two reviewer comments (RC1 and RC2). The two reviewer comments (and their supplemental files) were similar, but not identical in content. In neither reviewer comment does the reviewer suggest changes to the manuscript. Because no changes were suggested, no changes were made.

**Reviewer Comment 3 (Referee #2, Anonymous)**

Referee #2 provide multiple suggestions to improve the manuscript. These suggestions are pasted below. A description of how we have addressed the comments in a revision of the manuscript is provided in *italic*.

This paper contributes in reliable forecasts of building damage caused by debris flows. Three different models to represent a debris flow event are used, RAMMS, FLO-2D, and D-Claw. The paper is well structured and written in general. The recommendation is to accept it for publishing. I recommend handling the comments given below. And, it is very important to clarify the meaning of the term "hv^2", as pointed out below.

*We thank the reviewer for undertaking the review and for the below comments. We agree that addressing each of them will improve the quality and clarity of the manuscript. We also agree that it is important to clarify the meaning of the term 'hv$^2$'. We describe how we have addressed that issue in response to comment #6 below.*

1. The redaction style of the abstract must be improved. The word "forecasts" is appearing plenty of times, to mention one of the issues. It is a key word, however, redaction can be improved with no necessity of recurring (using) to synonyms.

*In the original text, we used 'forecast' as both a noun, a verb, and an adjective. We agree that this word is used many times, but this is a consequence of the manuscript being about forecasts. We concluded that it was most important that the usage of 'forecast' as a noun remain unchanged because it is this usage where precision of language is important (we do not want to use two words to mean that same thing). Therefore, we worked through the entire document and changed all instances of the word 'forecast' used as a verb to 'predicted'. We also evaluate each use of the word 'forecast' as a noun and removed those that we determined could be removed. This still left many instances where 'forecast' is used as a noun. We elected to retain the use of 'forecast' as an adjective because in these cases we are typically discussing a product of a forecast. However, we think that not using "forecast" as a verb is a substantial improvement and appreciate that the reviewer's comment forced us to evaluate the use of language.*

*To address the reviewer's general concern about overuse of words in the manuscript, we conducted a word frequency analysis to identify other words that we used many times in the text (Figure 1). We concluded that the other words we used frequently (e.g., 'building', 'damage') all have precise meaning in the context of this manuscript. Therefore, we did not make additional word choice changes.*

[Figure]

*Figure 1. Count of words in main and supplementary text after words were lemmatized using NLTK WordNet Lemmatizer (https://www.nltk.org/api/nltk.stem.wordnet.html#nltk.stem.wordnet.WordNetLemmatizer). Note on terminology: in linguistics, a 'lemma' is the canonical version of a word; for example, including all inflected forms of a verb. The lemmatizer was used to group together related words such that "forecast" and "forecasts" are counted together.*

2. L76-L80. Improve redaction.

*We have revised this paragraph to improve the clarity, including removing one of the uses of 'forecast'.*

3. L82-L83. These two paragraphs might be better unified. The description of the sections could be also better performed mentioning even those more specific procedures with no necessity of referencing every single subsection title.

*We have revised these two paragraphs to unify them. We have also removed the reference to subsection titles.*

4. L103-L104. Improve redaction.

*The original text of the sentence the reviewer is referring to is:*

*"Prior work estimated the total amount of sediment deposited in the event (Kean et al., 2019b), eroded from the hillslopes (Alessio et al., 2021), and eroded from the channels (Morell et al., 2021)."*

*We were not sure how to improve the clarity of this sentence because the sediment budget for this event was constrained by three separate studies each focused on a different aspect of the budget (deposit, hillslope, and channel). Because the reviewer was not more specific about how this text could be improved, we were unsure as to how to proceed. We did think that a slight modification to the first phrase would improve the text and have changed it to the following (addition underlined):*

*"Prior work estimated the total amount of sediment deposited in low sloping areas in the event (Kean et al., 2019b), eroded from the hillslopes (Alessio et al., 2021), and eroded from the channels (Morell et al., 2021)."*

5. Figure 1. Why the simulation domains (or boundaries of these three creeks) were not demarked or selected by using the watershed divide? Avoiding lack spaces between the creeks and event worst overlapping the domains with no necessity.

*We determined the simulation domains to extend well beyond the area impacted by debris flow runout associated with each creek. We did this because watershed divides are harder to determine in divergent topography and because debris flow runout may cross watershed divides. We have added a sentence to the main text to clarify this point This portion of the next now read (with new text underlined)*

*Finally, Barnhart et al. (2021) split up the complex runout path from the Montecito event into three independent simulation domains for the purpose of computational efficiency (Figure 1). The extent of each domain was drawn to encompass a region that is larger than the runout associated with each of the three major creeks (Montecito Creek, San Ysidro Creek, and Romero Creek).*

6. "hv^2" is in fact representing the momentum flux, as it is coming from the depth-averaged momentum equation (see FEMA, 2022a, p. 5-28). Correct this for the entire manuscript. Also see and cite:

  * Tan, W. Y. (1992). Shallow water hydrodynamics: Mathematical theory and numerical solution for a two-dimensional system of shallow-water equations. Elsevier.

  * Vreugdenhil, C. B. (1994). Numerical methods for shallow-water flow (Vol. 13). Springer Science & Business Media.

*We agree with the reviewer that the term $hv^2$ represents the momentum flux and is commonly called the momentum flux within the shallow water equations. However, the term itself is not a momentum flux (as it does not have units of momentum per unit area per time). Instead, it is the momentum flux per unit cross section per density.*

*We are glad the reviewer raised this issue because the reviewer is correct that we should not have called this a volume flux. We have changed all references of 'volume flux' in the manuscript to 'momentum flux' and added the following note to the end of the paragraph where momentum flux is introduced. We thought that this note was the correct place to include the suggested references, which we agree are appropriate to add to support this clarification.*

> *"(Note: The quantity $hv^2$ does not have units of a momentum flux ($kg\ m^{-1}\ s^{-2}$) but is called the momentum flux because within the shallow water equations $hv^2$ represents the transport flux of the momentum density, hv [e.g., Tan, 1992; Vreugdenhil, 1994]).”*

1. Could not be better write "100 by"? One more parenthesis is needed to close the complementary text.

*We have corrected both items identified by the reviewer.*

2. Figure 3. Make it clear if Figure 3a, is also a product of your paper or is this was taken from other document.

*The entirety of Figure 3 was drafted by the authors for the purpose of the paper. The general methodology of using a hazard intensity model to generate a forecast damage state (central portion of Figure 3a) is not novel as it is commonly used for making damage forecasts. We are not aware of examples for debris-flow runout forecasts that separately evaluate the role of boundary and initial conditions (left portion of Figure 3a). We produced the general form of the approach to indicate that it is more general than the specific version we implemented (with specific models and specific hazard intensities).*

*Because Figure 3a predominantly reflects original work, we have made no changes.*

3. L248 and L250. "Dimensionless" is more appropriated instead of "unitless". And, explain better meaning of K_D.

*We have revised the term 'unitless' to 'dimensionless throughout the manuscript.*

*Regarding the term $K_D$, we have expanded the definition of $K_D$ to now state:*

> *"$K_D$ (dimensionless) accounts for uncertainty in loading (e.g., $K_D<1$ to account for the effect of shielding or $K_D>1$ for the impact of individual boulders entrained in the flow, FEMA, 2022a p.5-28),”*

4. Define more appropriately the definition of the density used for your work. Is it the weighted averaged density of both water and soil particles?

*In the final paragraph of Section 4.2.2 we state*

> *"We used a debris-flow density of 2020 $kg\ m^{-3}$ reflecting a weighted average of water (1000 $kg\ m^{-3}$) and sediment (2700 $kg\ m^{-3}$) and a solid volume concentration of 0.6.”*

*We expect that this comment indicates that the reviewer was not certain that this is the value we used for the flow density in Equation (3). To make sure that this is clear we now write:*

*"For the density of the flow, r, we used a debris-flow density of 2020 kg m$^{-3}$ reflecting a weighted average of water (1000 kg m$^{-3}$) and sediment (2700 kg m$^{-3}$) and a solid volume concentration of 0.6."*

5. Are you estimating or assuming that hv^2=2/3?

*The assumption that the median value of $hv^2$ is equal to 2/3 of the maximum value of $hv^2$ comes directly from the FEMA Hazus method. To emphasize that this is an existing method we changed the word 'estimated' to 'calculated'. The sentence that states this reads as follows:*

*Following the Hazus methodology for estimating building damage based on tsunami flow, we calculated $\overline{hv^2}$ as 2/3 of the peak momentum flux (FEMA, 2022a, p.4–18).*

*Page 4-18 in the Hazus documentation (FEMA, 2022a) explains the basis for this assumption.*

6. 5. Correct B_w.

*The subscript W was added to B in equation 5. Thank you for catching this omission.*

7. "We discuss the implications of these simplifications later in Section ---."

*We now refer to the section where these simplifications are discussed.*